



# Measurement report: Cloud Processes and the Transport of Biological Emissions Regulate Southern Ocean Particle and Cloud Condensation Nuclei Concentrations

Kevin J. Sanchez[1*], Gregory C. Roberts[1,2], Georges Saliba[1], Lynn M. Russell[1], Cynthia Twohy[3], Michael J. Reeves[4], Ruhi S. Humphries[5], Melita D. Keywood[5], Jason P. Ward[5], Ian M. McRobert[6]

[1]Scripps Institution of Oceanography, University of California, San Diego, CA, USA
[2]Centre National de Recherches Météorologiques, Météo-France & CNRS UMR3589, Toulouse, France
[3]NorthWest Research Associates, Redmond, WA, USA
[4]National Center for Atmospheric Research, Boulder, CO, USA
[5]Climate Science Centre, CSIRO Oceans and Atmosphere, Aspendale, Australia
[6]Engineering and Technology Program, CSIRO Oceans and Atmosphere, Hobart, Australia

[*]Now at: Universities Space Research Association, Columbia, MD, USA
 Now at: NASA Langley Research Center, Hampton, VA, USA

*Correspondence to*: Kevin J. Sanchez (kjs356@gmail.com)

**Abstract.** Long-range transport of biogenic emissions from the coast of Antarctica, precipitation scavenging, and cloud processing are the main processes that influence the observed variability in Southern Ocean (SO) marine boundary layer (MBL) condensation nuclei (CN) and cloud condensation nuclei (CCN) concentrations during the austral summer. Airborne particle measurements on the HIAPER GV from north-south transects between Hobart, Tasmania and 62°S during the Southern Ocean Clouds, Radiation Aerosol Transport Experimental Study (SOCRATES) were separated into four regimes comprising combinations of high and low concentrations of CCN and CN. In 5-day HYSPLIT back trajectories, air parcels with elevated CCN concentrations were almost always shown to have crossed the Antarctic coast, a location with elevated phytoplankton emissions relative to the rest of the SO. The presence of high CCN concentrations was also consistent with high cloud fractions over their trajectory, suggesting there was substantial growth of biogenically formed particles through cloud processing. Cases with low cloud fraction, due to the presence of cumulus clouds, had high CN concentrations, consistent with previously reported new particle formation in cumulus outflow regions. Measurements associated with elevated precipitation during the previous 1.5-days of their trajectory had low CCN concentrations indicating CCN were effectively scavenged by precipitation. A course-mode fitting algorithm was used to determine the primary marine aerosol (PMA) contribution which accounted for < 20% of CCN (at 0.3% supersaturation) and cloud droplet number concentrations. Vertical profiles of CN and large particle concentrations ($D_p > 0.07 \mu m$) indicated that particle formation occurs more frequently above the MBL; however, the growth of recently formed particles typically occurs in the MBL, consistent with cloud processing and the condensation of volatile compound oxidation products.





CCN measurements on the *R/V Investigator* as part of the second Clouds, Aerosols, Precipitation, Radiation and atmospheric Composition Over the southeRn Ocean (CAPRICORN-2) campaign were also conducted during the same period as the SOCRATES study. The *R/V Investigator* observed elevated CCN concentrations near Australia, likely due to continental and coastal biogenic emissions. The Antarctic coastal source of CCN from the south as well as CCN sources from the mid-latitudes create a latitudinal gradient in CCN concentration with an observed minimum in the SO between 55°S and 60°S. The SOCRATES airborne measurements are not influenced by Australian continental emissions, but still show evidence of elevated CCN concentrations to the south of 60°S, consistent with biogenic coastal emissions. In addition, a latitudinal gradient in the particle composition is observed; more hygroscopic particles to the north, consistent with a greater fraction of sea salt from PMA, and more sulfate and organic particles to the south, which are likely from biogenic sources in coastal Antarctica.

## 1 Introduction

The marine boundary layer (MBL) over the Southern Ocean (SO) displays some of the most pristine conditions in the world with few anthropogenic influences, making cloud properties and radiative forcing particularly sensitive to relatively small changes in aerosol source emissions (Downey et al., 1990; Fossum et al., 2018; Hudson et al., 1998; Li et al., 2018; McCoy et al., 2015; Murphy et al., 1998b; Pandis et al., 1994; Pierce and Adams, 2006; Pringle et al., 2009; Whittlestone and Zahorowski, 1998; Wood et al., 2015; Yoon and Brimblecombe, 2002). In spite of a growing number of studies, climate models still struggle to represent SO cloud radiative properties, partly because their representation of available cloud condensation nuclei (CCN) is not well constrained (Bodas-Salcedo et al., 2014; Brient et al., 2019; Carslaw et al., 2013; Hyder et al., 2018; Lee et al., 2015; Mace and Protat, 2018; McCoy et al., 2014; Ogunro et al., 2018; Schmale, 2019; Trenberth and Fasullo, 2010). A major shortcoming in the ability of Global Climate Models (GCM) to predict future climate change (i.e., insufficient cloud forcing (Bodas-Salcedo et al., 2014)) is associated with biases in satellite measurements and a lack of in-situ measurements needed to validate satellite and models in the SO (Lenschow et al., 1999; Seinfeld et al., 2016). Understanding the impact of these particle sources on the cloud system and their variability is required for accurate prediction of SO cloud properties and to understand the impact of aerosol-cloud interactions on the Earth's energy budget. These issues motivated the Southern Ocean Clouds, Radiation Aerosol Transport Experimental Study (SOCRATES), which involved in-situ measurements of clouds and aerosol over the SO on-board the NSF/NCAR HIAPER Gulfstream V (GV) (UCAR/NCAR - Earth Observing Laboratory, 2005) . The analysis covered in this study identifies patterns associated with variation in the observed condensation nuclei (CN) and CCN concentrations over the SO.

Aerosol concentrations in the SO are typically derived from natural marine sources and rarely influenced by continental or anthropogenic sources. These marine sources consist of primary marine aerosol (PMA) particles produced from sea spray and bubble bursting, and secondary organic and sulfate particles formed from biologically emitted volatile organic compounds (VOCs) such as dimethyl sulfide (DMS) (Bates et al., 1998b, 2012; Covert et al., 1992; Frossard et al., 2014; Middlebrook et al., 1998; Murphy et al., 1998a; Pirjola et al., 2000; Quinn et al., 2000, 2017; Rinaldi et al., 2010; Saliba et al., 2019). Primary





particles from the Antarctic continent are not a major source of particles to the SO because it is mostly covered in ice (Chambers et al., 2017); so the main sources of primary aerosol from Antarctica to the SO are limited to local anthropogenic pollution from research stations, blowing snow, frost flowers and sea bird emissions (Frieß et al., 2004; Huang and Jaeglé, 2017; Liu et al., 2018; Schmale et al., 2013).

New particle formation (NPF) from the oxidation of marine biologically emitted VOCs occurs when the particle condensational sink is low and temperature is low, both of which are prevalent conditions over the SO (Raes et al., 1997; Yue and Deepak, 1982). While new particle formation has been observed in the SO marine boundary layer (Covert et al., 1992; Humphries et al., 2015; Kyrö et al., 2013; Pirjola et al., 2000; Weller et al., 2015) it occurs more commonly in the free troposphere (Bates et al., 1998b; Clarke et al., 1998; Humphries et al., 2016; Odowd et al., 1997; Reus et al., 2000; Sanchez et al., 2018; Yoon and

Brimblecombe, 2002) owing to the absence of PMA in the SO MBL, which reduces total particle surface area (McCoy et al., 2015). In addition, new particle formation is commonly associated with cumulus outflow regions due to the DMS-rich air lofted by the convection and the high relative humidity, creating an environment that may be able to allow binary nucleation between sulphuric acid (a DMS oxidation product) and water (Bates et al., 1998b; Clarke et al., 1999; Cotton et al., 1995; Perry and Hobbs, 1994; Twohy et al., 2002), but ternary nucleation with ammonia or amines is also possible (Weber et al.,

1998).

Furthermore, the remote mid-latitude SO contains much less biological activity near the ocean surface relative to the Antarctic continental coast, which creates a latitudinal gradient in aerosol concentrations driven by biogenic particle formation (Alroe et al., 2019; Humphries et al., 2016; Kim et al., 2019; O'shea et al., 2017; Odowd et al., 1997; Weller et al., 2018). This trend in biology is linked to observations showing a distinct transition in aerosol properties around 64°S where CN concentrations

south of 64°S are about five times higher during the austral spring months (Alroe et al., 2019; Humphries et al., 2016). Regions of sea ice melt on the Antarctic coast have been observed to be a significant source of methanesulfonic acid (MSA) as well as DMS and organic nitrogen (Dallósto et al., 2017; Dowd et al., 1997; Vana et al., 2007), known precursors of new particle formation (Almeida et al., 2013; Dawson et al., 2012). For this reason, long-range transport of CN and gaseous precursors in the MBL and free troposphere from the Antarctic continental coast plays a significant role in enhancing CN and CCN

concentrations in the SO (Bates et al., 1998a; Clarke et al., 1998, 2013; Dzepina et al., 2015; Korhonen et al., 2008; Woodhouse et al., 2010).

The enhanced CN concentrations from biogenic sources are transported from coastal Antarctica to the remote SO, suggesting that biological particles may account for a significant fraction of SO particles. Other field studies have shown large variability in the contribution of PMA particles to total CN concentrations, ranging between 8-100% while the remaining 0-92% is from

biogenic organic and sulfate particles (Fossum et al., 2018; Gras and Keywood, 2017; McCoy et al., 2015; Quinn et al., 2017; Yoon and Brimblecombe, 2002). Much of the observed variability is linked to seasonal variations in SO biological activity (Ayers and Gras, 1991; Korhonen et al., 2008). On the Antarctic peninsula, NPF events occurred mostly during the austral summer, with CCN concentrations (at 0.4% supersaturation) increasing on average by 11% (Kim et al., 2019). Similarly, higher average concentrations of cloud droplet number concentrations (CDNC) are observed in the austral summer (Mace and





Avey, 2017; McCoy et al., 2015). Conversely, PMA CCN is found to have little seasonal variability relative to biogenic CCN (Vallina et al., 2006), likely driven by small seasonal differences in wind speed (Saliba et al., 2019). However, though some studies suggest biologically productive waters enhance PMA production, which would also increase CCN concentrations in coastal biologically productive regions (Collins et al., 2016; Fuentes et al., 2010). In a recent study, a parcel model showed that PMA may limit CCN concentration indirectly, by enhancing in-cloud water uptake at low supersaturations, causing a decrease of maximum in-cloud supersaturations and cloud droplet number concentrations by suppressing the activation of smaller particles (Fossum et al., 2020). Consequently, an increased fraction of small particles remain as interstitial aerosol and do not undergo growth via cloud processing. Likewise, previous observations of wind speed and the production of PMA negatively correlate with accumulation mode particle concentrations (Gras and Keywood, 2017). Furthermore, organic enrichment of PMA in biologically productive waters can further reduce their hygroscopicity (Burrows et al., 2018; Cravigan et al., 2019; Law et al., 2017; Meskhidze and Nenes, 2010).

With substantial growth of newly formed particles by the uptake of VOC oxidation products through cloud processing, biogenic sourced particles may grow CCN larger and increase CDNC (Hoppel et al., 1986; Hudson et al., 2015; Pirjola et al., 2004; Russell et al., 2007; Sanchez et al., 2018). Cloud processing occurs when small particles activate to form cloud droplets, leading to enhanced condensation of VOC oxidation products onto the droplet because the droplet surface area is larger than that of the unactivated particles. Aqueous phase oxidation of absorbed VOCs also results in the formation of less volatile compounds, which remain in the particle phase upon evaporation of the water (Hoppel et al., 1986). In the event that the cloud droplets do not precipitate, the evaporated particles are larger than their original size since non-volatile compounds (i.e., sulfate and MSA) that condensed onto the cloud droplet remain in the particle phase. This added mass shifts Aitken mode particles to the accumulation mode (Hoppel et al., 1986; Hudson et al., 2015; Kaufman and Tanré, 1994; Sanchez et al., 2017). Results from McCoy et al. (2015) emphasized this sensitivity and showed global model simulations of natural aerosol account for more than half the spatial and temporal variability in the satellite derived CDNC over the SO. These areas of enhanced CDNC also correlate with areas of high chlorophyll-a, a tracer for phytoplankton activity, increasing secondary sulfate and organic aerosol concentrations (Krüger and Graßl, 2011; McCoy et al., 2015). In contrast, the inclusion of PMA in a GCM increased CCN concentrations at 0.2% supersaturation ($CCN_{0.2}$) by 150-500% (Pierce and Adams, 2006), suggesting PMA is the largest contributor to CCN concentrations. SO satellite-derived cloud properties (liquid water content (LWC), effective radius, cloud fraction) showed seasonal variations that resulted in a difference in cloud radiative forcing (i.e., surface cooling) between 14 and 23 W m$^{-2}$ (McCoy et al., 2014). Increased CDNC is also shown to correlate with enhanced cloud fraction, significantly increasing overall cloud shortwave forcing (Rosenfeld et al., 2019). If cloud droplets do not evaporate and instead precipitate, CN and CCN concentrations are reduced through precipitation scavenging (Croft et al., 2010; Stevens and Feingold, 2009).

In this study, we discuss airborne CN and CCN measurements from the SOCRATES campaign. Measurements are divided into four categories based on the total CN and $CCN_{0.3}$ (CCN concentration at 0.3% supersaturation) to identify differences in processes and sources that lead to the observed variability of measurements. A back trajectory analysis is performed to identify the source of air parcels and their history with respect to their proximity to clouds and precipitation. Additionally, a PMA



mode fitting algorithm (Saliba et al., 2019) is utilized to understand the contribution of PMA to CCN and observed CDNC

concentrations. The findings describe the observed spatial gradients and relative importance of biogenic sulfate and PMA to

CDNCs, which ultimately contribute to improving estimates of the energy budget in the SO.

## 2 Methods

### 2.1 NSF/NCAR HIAPER GV measurements and *R/V investigator* CCN measurements

Airborne measurements were collected on the NSF/NCAR Gulfstream-V High-performance Instrumented Airborne Platform

for Environmental Research (GV HIAPER) observational platform. The GV was stationed at the Hobart International Airport,

Tasmania, during the austral summer between 15 January and 24 February 2018. The flight strategy during SOCRATES

involved ferrying out to a predetermined area of interest followed by a series of straight vertical profiles, and level legs to

sample below, in and above clouds. The GV HIAPER conducted 15 research flights (RF) over the SO between 42.5 °S and

62.1 °S and between 133.8 °E and 163.1 °E at altitudes ranging from 50-7500 m. Flight tracks and flight strategy are presented

in McFarquhar et al. (submitted) Figure 1 and 4 respectively.

A wing-mounted Ultra-High Sensitivity Aerosol Spectrometer (UHSAS, Droplet Measurement Technologies, Boulder, CO)

measured particle size distribution between 0.06 and 1.0 μm in diameter; however, the 0.06-0.07 μm diameter range was not

used in this analysis due to instrument noise. Ambient subsaturated particles collected with the UHSAS were dried through

ram heating and deiced components. A condensation particle counter (CPC, TSI 3760A) was used to measure total particle

concentrations (CN, diameter > 0.01 μm). CCN measurements were performed with two miniature continuous-flow stream-

wise thermal gradient chambers, one in scanning supersaturation mode and one in constant supersaturation mode (Roberts and

Nenes, 2005). The supersaturation range of the scanning CCN counter spanned from 0.06 % to 0.87 % and recorded a

continuous CCN spectrum every five minutes. The constant supersaturation CCN counter operated at 0.43 % supersaturation

(referred to as $CCN_{0.43}$), at 1 Hz and was used to identify CCN gradients in vertical profiles (Section 3.6). CCN concentration

at 0.3% supersaturation ($CCN_{0.3}$, derived from the scanning CCN counter) was used throughout this study as a reference CCN

concentration because $CCN_{0.3}$ best corresponded to observed CDNCs (Section 3.3). The internal chamber pressure of both

CCN counters was controlled to 400 hPa. A cloud droplet probe (CDP, DMT, Boulder, CO) was used to measure cloud droplet

concentration (2-50 μm wet diameter). The scanning CCN spectra and UHSAS number size distributions were used to estimate

the 0.07 μm diameter hygroscopicity parameter ($\kappa_{70}$) for each MBL leg. All particle measurements were converted to surface

standard temperature and pressure. CN and CCN measurements made in-cloud (defined by CDP measurements of LWC > 0.1

g kg$^{-1}$) were excluded from the analysis due to the influence of droplet shattering within the aerosol inlets. During the research

flights, areas of intense precipitation were avoided, but some measurements were made under drizzle and light rain conditions;

however, there was no evidence of droplet shattering in the inlets under these conditions.

In addition to the SOCRATES GV HIAPER measurements, the *R/V Investigator* (CSIRO, Hobart, Tasmania) also collected

aerosol and sea water samples during the second Clouds, Aerosols, Precipitation, Radiation and atmospheric Composition





Over the southeRn Ocean (CAPRICORN-2) campaign. The CAPRICORN-2 study was conducted from 10 January to 21 February 2018, overlapping the SOCRATES study. The R/V Investigator covered a north-south transect over the SO, starting at Hobart, Tasmania (43°S) and reaching approximately 66°S, and then returning to Hobart. In this study, CCN measurements collected on the *R/V Investigator* were measured with a commercially-available streamwise CCN counter (CCN-100, Droplet

Measurement Technologies, Boulder, CO) that measured CCN concentration between 0.25 and 1.05% supersaturation with a step-wise scan. Each CCN spectrum took approximately one hour to complete. *R/V Investigator* CCN at 0.35% are analyzed and compared to the GV HIAPER $CCN_{0.3}$ measurements.

### 2.2 Model data

#### 2.2.1 HYSPLIT-GDAS

In this study, HYSPLIT hourly five-day back trajectories were performed with the Global Data Assimilation System (GDAS, ftp://arlftp.arlhq.noaa.gov/pub/archives/gdas0p5/) (Rolph et al., 2017; Stein et al., 2015) at 0.5 degree resolution for each CCN spectrum in the MBL leg (below the cloud layer if clouds are present). The average latitude, longitude and altitude (50-500 m) of the MBL legs on the GV HIAPER were used as starting points for the back trajectories. Antarctica is the only continent over which back trajectories passed; none of the airborne aerosol measurements in the MBL were influenced by continental

Australia; the only anthropogenic influences were potentially ship tracks and research stations in Antarctica, which we assume to have a negligible impact in this study.

#### 2.2.2 ECMWF ReAnalysis (ERA5)

ERA5 is the 5th generation of a climate reanalysis dataset from the European Centre for Medium-Range Weather Forecasts (ECMWF)(Copernicus Climate Change Service (C3S), 2017). The ERA5 model assimilates satellite, ground and airborne

measurements to archive the state of the weather and climate. The ERA5 total precipitation and low-level cloud fraction was used for the time period covering the SOCRATES campaign to identify the role of clouds and precipitation in changing CN and CCN concentrations. The ERA5 time resolution is hourly, and spatial resolution is 0.25 degrees.

### 2.3 Primary Marine Aerosol (PMA) Fitting Algorithm

The PMA concentration was determined by fitting the UHSAS distribution of particles greater than 0.2 μm diameter to a single

lognormal mode. A single lognormal mode has been found to represent PMA in ambient measurements (Modini et al., 2015; Quinn et al., 2017; Saliba et al., 2019). While this method was previously used on dry particle number size distributions ranging from 0.02 to 5.0 μm (Saliba et al., 2019), the UHSAS measures the particle number size distribution between 0.07 and 1.0 μm diameter. In addition, UHSAS particles in the SOCRATES campaign were not fully dried and a relatively narrow deliquesced mode (GSD =1.44±0.25) is present at approximately 0.6 μm diameter, similar to previous measurements of optically derived

particle distributions at high relative humidity (Strapp et al., 1992). This 0.6 μm deliquescent mode was consistently fit by the algorithm. The deliquesced UHSAS particles affect the mode diameter of the fitted PMA size distribution but not the retrieved



PMA (and $CCN_{0.3}$) number concentrations. The concentration of particles in this fitted mode correlated moderately with wind speed (Section 3.5), similar to previous measurements of PMA estimated with this method (Modini et al., 2015; Quinn et al., 2017; Saliba et al., 2019), indicating the fitted mode is a viable approximation of PMA concentrations. The estimated PMA
mode sizes are consistent with sea salt (from PMA) observed on size-resolved particles collected in the marine boundary layer during SOCRATES and analysed with transmission electron microscopy (TEM). The TEM analysis shows almost all particles > 0.2 µm in diameter consist of sea salt; and sea salt particles account for 25% to 100% of particle number concentrations at particle diameters > 0.4 µm (Saliba et al. submitted).

## 3 Results

### 3.1 Particle Regimes

MBL CN and $CCN_{0.3}$ measured on the GV HIAPER MBL legs ranged from 116-1153 $cm^{-3}$ and 17-264 $cm^{-3}$ and averaged 540±246 $cm^{-3}$ and 123±58 $cm^{-3}$, respectively. Figure 1b shows the CN and $CCN_{0.3}$ concentrations averaged over each CCN spectra scan during GV HIAPER MBL legs throughout the SOCRATES field project (with the exception of RF 14 when the scanning CCN counter malfunctioned). To determine which atmospheric processes drove the variability of nearly an order of
magnitude in CN and $CCN_{0.3}$, the measurements were divided into four regimes. The regime thresholds were selected based on the bimodality of observed CN and $CCN_{0.3}$ concentrations shown by the histograms and kernel density functions in Figure 1a,c. Figure 1a,c also shows the kernel density estimate based on a normal kernel function. Using this approach, rather than grouping all values into a single bin, each measurement is represented by a normal distribution and integrated to produce the kernel density estimate. The optimal kernel density estimate bandwidth was found to be 28 and 91 for $CCN_{0.3}$ and CN,
respectively, and calculated using the "ksdensity" function from Matlab (2019), derived from theory developed by Bowman and Azzalini (1997). The Hartigan's Dip test (Hartigan and Hartigan, 1985) determined that the distribution was not unimodal (P-value <0.01) for both CN and $CCN_{0.3}$, thereby validating the use of a bimodal distribution for this analysis. The bimodal distribution minima correspond to 125 $cm^{-3}$ and 750 $cm^{-3}$ for $CCN_{0.3}$ and CN, respectively. Even though only $CCN_{0.3}$ was used to determine the particle regimes, Figure 2a illustrates the systematic differences between the averaged CCN spectra and CN
concentrations for each of the regimes. The bimodal $CCN_{0.3}$ and CN regimes were combined for a total of four regimes. Table 1 shows the average $CCN_{0.3}$ and CN concentrations for each of the four regimes, which are distinguished by permutations of high and low $CCN_{0.3}$ and CN concentrations.

To differentiate the four regimes in the text, we have given them each abbreviated descriptive names based on their CN and $CCN_{0.3}$ concentrations, where the regime with high CN and CCN concentrations is referred to as "Recent particle formation
(RPF) + Aged", the regime with low CN and CCN concentrations is referred to as "Scavenged", the regime with low CN and high CCN concentrations is referred to as "Aged", and finally the regime with high CN and low CCN concentrations is referred to as "RPF + Scavenged". The classification of each regime is based on the relative concentration of Aitken-mode particles (CN) and accumulation-mode particles (CCN), with a naming convention that describes the expected airmass history. Similar



to analyses in previous studies, the relative contribution of the accumulation-mode and the Aitken-mode are used to identify
recent particle formation (RPF) events and growth of Aitken-mode particles to accumulation-mode or CCN sizes (Kalivitis et
al., 2015; Kleinman et al., 2012; Williamson et al., 2019). The Scavenged regime is named based on evidence indicating the
removal of CCN-sized particles through precipitation scavenging (Section 3.3). The Aged regime represents cases in which
accumulation-mode is prominent and CCN particle concentrations are relatively high, likely due to atmospheric processes that
increase particle size over time such as the condensation of VOC oxidation products or cloud processing (Section 3.2 and 3.3,
respectively). The RPF regime exhibits a large Aitken-mode with high concentrations of CN, indicative of recent particle
formation (Section 3.2).

### 3.2 Back Trajectories

Previous studies have shown long-range transport of particles and VOCs can affect locally observed particle concentrations
and chemical properties (Dzepina et al., 2015; Korhonen et al., 2008). In addition, atmospheric processes affecting particle
concentrations upstream of the measurement location reduces the correlation of particle properties to individual (or discrete)
processes, such as precipitation, cloud processing, and new particle formation (Albrecht, 1989; Bates et al., 1998b; Russell et
al., 2009; Sanchez et al., 2018; Stevens and Feingold, 2009; Stevens and Seifert, 2008; Vallina et al., 2006; Wood et al., 2015).
Lagrangian HYSPLIT back trajectories initiated at MBL leg altitudes (50-500 m) were used to determine the path travelled by
the parcel of air for the previous five days for each of the MBL legs (Figure 3). Consistent patterns are apparent for each of
the particle regimes. Specifically, the back trajectories for the Aged particle regime (Figure 3d) are consistently from the south
along the Antarctic coast, which is associated with the elevated ocean surface emissions of DMS and other VOCs produced
by phytoplankton activity (Alroe et al., 2019; Humphries et al., 2016; Kim et al., 2019; O'shea et al., 2017; Odowd et al., 1997;
Weller et al., 2018). In contrast, the high CN regimes (RPF and RPF + Scavenged) exhibit back trajectories generally from the
west from the SO. The scavenged regime consists of back trajectories from both the west and the south, signifying atmospheric
process rather than the parcel path and origin influence the observed CN and CCN concentrations.

### 3.3 Cloud Processing

Relating the identified regimes to the observed cloud processes provides insight on how cloud processes affect CN and CCN
concentrations. Figure 2c shows $CCN_{0.3}$ and CDNC correlated moderately (r = 0.75), the highest correlation of CCN
concentrations relative to other supersaturations, indicating that $CCN_{0.3}$ is a good proxy for CDNC, and similar to previous
estimates of marine cloud effective supersaturations (Martin et al., 1994; Snider et al., 2003). For this comparison, the 90th
percentile of CDNC from vertical profiles are matched to the nearest below-cloud MBL leg CCN concentration. As expected,
the Aged particle regime accounted for cases with the highest CDNCs (192±100), while the scavenged particle regime
accounted for the lowest observed CDNC (111±72). Few CDNC measurements are associated with the RPF (high CN) regimes,
suggesting fewer clouds are associated with this regime. Figure 2b shows the cloud effective supersaturation and its



relationship to the CDNC. The cloud effective supersaturation is calculated as the supersaturation where the CCN concentration was equal to the 90th percentile of the measured CDNC. Typically, clouds contain a range of peak supersaturations, controlled by the strength of the updraft and the cloud droplet number concentration (Hudson and Svensson, 1995; Pawlowska and Grabowski, 2006; Siebert and Shaw, 2017). The effective supersaturation accounts for the CCN that
have activated adiabatically near cloud base and subsequently dried through sub-adiabatic mixing processes (Sanchez et al., 2017). In general, the observed CDNC weakly correlate to the effective supersaturation (Figure 2b, r = 0.47). The two regimes with aged particles (high CCN) consistently had higher CDNCs than the other regimes, highlighting the role of CCN concentrations on impacting CDNC. It is also important to note, CDNC can still be relatively high (~200 cm$^{-3}$) in regimes with low CCN under conditions of high in-cloud supersaturations generated by strong updrafts, or with relatively low PMA
concentrations which also allows the generation of higher in-cloud supersaturations (Fossum et al., 2020).

To identify the effect of precipitation on CCN concentrations, $CCN_{0.3}$ is compared to the total precipitation (obtained from ERA5) integrated over a 35-hour back trajectory, shown in Figure 4a. As expected, the two scavenged regimes (with lower $CCN_{0.3}$ concentrations) corresponded to higher total precipitation. Figure 4b shows the Pearson correlation coefficient comparing the base 10 logarithm of the integrated total precipitation over back trajectory times of 0 to 120 hours and CCN
concentrations between 0.1% and 0.8% supersaturation. The Pearson's coefficient r-value peaked for 35-hour back trajectories at CCN supersaturations ranging from 0.3-0.5% (similar to effective in-cloud supersaturations, Figure 4b), indicating air parcel history, particularly in the last 1.5 days, is important for determining atmospheric processes that affect CCN concentration. The Pearson's coefficient for $CCN_{0.1}$ was consistently the lowest, likely because $CCN_{0.1}$ is associated with PMA, which is quickly replenished in the MBL through sea spray emissions. Similarly, the Pearson's coefficient for $CCN_{0.87}$ was also low,
likely because this CCN size is associated with RPF particles that are replenished in the FT and subsequently grow to larger sizes (and lower supersaturation CCN).

Figure 4c shows the MBL cloud fraction (obtained from ERA5) over the 120-hour back trajectory averaged for each particle regime. The two regimes with RPF (RPF and RPF + scavenged; high CN) are associated with lower cloud fraction (< 0.6), which suggests the presence of cumulus clouds. New particle formation has previously been observed in cumulus cloud
outflow regions (Bates et al., 1998b; Clarke et al., 1999; Cotton et al., 1995; Perry and Hobbs, 1994) and is likely the main source of CN in these RPF regimes. In contrast, the "Aged" particle regimes correspond to high MBL cloud fraction (> 0.6), which is consistent with stratus and stratocumulus clouds. Stratus and stratocumulus clouds typically include less precipitation, allowing more cloud processing of CN to CCN sizes (Flossmann and Wobrock, 2019; Hoppel et al., 1990; Hudson et al., 2015; Neubauer et al., 2014). In addition, the concentration of ultrafine particles ($D_P$ < 30 nm) also decreases through Brownian
scavenging of interstitial particles onto cloud droplets (Croft et al., 2010), so that higher cloud fractions further reduce CN concentrations. The back trajectories associated with the Aged regime (Figure 3d) typically originate from SO storm tracks to the south, which is consistent with the elevated cloud fraction shown in Figure 4c. The storm track frequency peaks around 60°S (Patoux et al., 2009), suggesting parcels of air entering the storm track from the south have also been influenced by





coastal Antarctic biogenic DMS and other VOC emissions, eventually leading to increases in CCN concentrations via cloud

processing and in the absence of precipitation. The trajectories associated with the RPF and the RPF + Aged regimes are typically from the west, and have fewer clouds. While these regimes have elevated CN concentrations, they are not linked to Antarctic coastal sources within the last 120 hours (Figure 3a,b). Long-range transport of aerosol particles and their precursors for more than five days is possible in the absence of major sinks (i.e., precipitation) (Feichter and Leisner, 2009). The existence of both Aged and RPF in the same regime suggests particles have experienced some cloud processing as well as input from a

recent particle formation event. The cloud fraction for the aged + RPF regime is significantly lower than the aged regime (Figure 4c).

### 3.4 Latitudinal Gradient

Both the airborne GV HIAPER and shipborne *R/V Investigator* measurements showed latitudinal (North-South) gradients in CCN concentrations, although differences in the sampling strategies between the two platforms does result in slight differences

in the observed latitudinal gradients (discussed in detail below). Both sets of measurements showed high CCN concentrations near Antarctica (Figure 5a-c) consistent with Antarctic coastal biological emissions as a source of aerosol precursors. Back trajectories (Figure 3d; Aged regime) show that long-range transport of these Antarctic coastal emissions generates elevated aerosol concentrations as far north as ~50°S, almost 2000 km away from the Antarctic coast. The Pearson's Coefficient comparing airborne $CCN_{0.3}$ measurements with latitude suggest there is not a significant correlation (r =-0.09; Figure 5b),

unless the particles that were transported 2000 km across the SO from the Antarctic coast are excluded (r = -0.26). Similarly, there is no significant trend in airborne CN ($D_p$ > 0.01 μm) with latitude (r = 0.16) even though previous studies have noted a distinct increase in CN concentrations near the Antarctic shelf at 64°S (Alroe et al., 2019; Humphries et al., 2016). In SOCRATES, however, airborne measurements on the GV HIAPER reached only 62.1°S and did not capture the expected distinct increase in CN concentrations above the Antarctic coastal areas.

The presence of a latitudinal gradient in aerosol concentrations ($D_p$ > 0.07 μm) and a weak gradient in the GV HIAPER CCN implies a north-south gradient in particle composition (i.e., hygroscopicity) across the SO. Figure 5d shows the hygroscopicity parameter (κ) for $D_p$ > 0.07 μm derived at each MBL leg. The smaller κ (less hygroscopic aerosol) at high latitudes is consistent with the aerosol particles originating from biogenic emissions which have lower hygroscopicity values (κ = 0.6-0.9 for sulfates and κ < 0.2 for organics) relative to sea salt (κ = 1.3) (Kreidenweis and Asa-Awuku, 2014; Petters and Kreidenweis, 2007).

As PMA (mostly comprised of sea salt) is present all over the SO, relatively high κ values (κ ~1.0 (Quinn et al., 2014)) are found north of ~ 55°S where there are fewer biologically-derived organic and sulfate particles. The latitudinal trend of decreasing κ (i.e., more hygroscopic chemical composition toward the lower latitudes) implies particles further south in the SO will need higher in-cloud supersaturations to activate particles of the same size compared to mid regions of the SO where there are fewer biologically derived particles. Alternatively, particle growth and aging enhances the particle's ability to be

CCN active even with a low hygroscopicity and initial small size. Despite the lower observed hygroscopicity of particles at



high latitudes based on the airborne measurements, there are a greater number of CCN available (Figure 5b) to increase cloud droplet number and potentially enhance cloud reflectivity at higher latitudes.

Measurements from the *R/V Investigator* during the CAPRICORN-2 study show minima in CCN concentrations around 60°S (Figure 5a), which corresponds to the maximum in SO storm track activity (Patoux et al., 2009); however, this minima in CCN
is not observed from the GV measurements and is not as pronounced in similar ship measurements at the same time (Humphries et al., in prep.). As expected, based on the GV measurements, there are elevated CCN concentrations to the south of 60°S related to biogenic emissions from the Antarctic coastal areas. There are also elevated CCN concentrations north of 50°S measured on the *R/V Investigator*, probably related to continental emissions from Australia, phytoplankton emissions near the Australian coast, and even long-range transport of Antarctic coastal emissions (Ayers and Gillett, 2000; Twohy et al.,
submitted). The different latitudinal trends in CCN observed by the GV HIAPER and *R/V Investigator* are likely a result of the different temporal and spatial sampling strategies between the aircraft and the ship. The GV transects were repeated 15 times over 40 days while avoiding actively precipitating clouds, and represent the CCN variability across the SO. The GV typically started MBL measurements south of 50°S, so the trend in CCN concentrations is not as apparent in the GV measurements compared to the CCN gradient measured on the *R/V Investigator*. The *R/V Investigator* transected the SO twice,
with each transect occurring over 20 days.

### 3.5 Primary Marine Aerosol (PMA)

To explore the contribution of marine sources to CCN and CDNC, PMA was estimated from the UHSAS distributions through fitting of the PMA mode using the algorithm from Saliba et al. (2019). The retrieved PMA concentrations varied between <1 and 25 cm$^{-3}$ with an average of 6±3 cm$^{-3}$. The mode diameter of the retrieved PMA number size distribution was 0.59 ± 0.04
µm, which is consistent with the average mode diameter observed in the North Atlantic of 0.54 ± 0.21 (Saliba et al., 2019). The variability for the retrieved mode diameter of the PMA number size distribution likely reflects the available statistics (N = 74), and the possibility that the UHSAS particles were not completely dry (section 2.3). The calculated PMA number concentrations moderately correlated to wind speed (r = 0.53, Figure 6a), as also reported by Saliba et al. (2019) over the North Atlantic and Bates et al. (1998b) south of Australia. Using the ratio of the PMA and CCN$_{0.3}$ concentration (Figure 6b), the
PMA contribution to SO clouds can be estimated. PMA accounts for up to ~20% of CCN$_{0.3}$ (and CDNC), even for conditions with precipitation scavenging in the previous 1.5 days (Figure 4a), and only a small fraction compared to the biogenically generated aerosol. These results are similar to Quinn et al. (2017) who found that PMA contributed to less than 30% of CCN (at 0.3% supersaturation) over the SO, and Twohy et al. (submitted) who found sea-spray aerosol comprised a minority of cloud droplet residual number in 3 SOCRATES cases. Others have reported higher contributions, however of > 50% and even
up to 100% at high wind speeds (>16 m s$^{-1}$) for supersaturations <= 0.3%, during the austral summer (Fossum et al., 2018; Yoon and Brimblecombe, 2002).

### 3.6 Vertical Transport



High concentrations of aerosol particles in the MBL can be formed during new particle formation events in the free troposphere and subsequently entrained downward into the MBL (Bates et al., 1998a; Clarke et al., 1996, 2013; Korhonen et al., 2008;
Pirjola et al., 2000; Reus et al., 2000; Russell et al., 1998; Sanchez et al., 2018; Thornton et al., 1997; Yoon and Brimblecombe, 2002). The nucleation of new aerosol particles often occurs in the free troposphere owing to the low total condensational sink and cold temperatures (Raes et al., 1997; Yue and Deepak, 1982). It has traditionally been thought that the SO is a possible exception to this trend because the SO MBL is a pristine environment with few anthropogenic sources, relatively low particle concentrations (condensational sink), and low temperatures compared to other MBLs around the world (Covert et al., 1992;
Humphries et al., 2015; Pirjola et al., 2000; Yue and Deepak, 1982). To determine if the SO MBL truly is an exception to the trend, we compare the concentrations of recently formed and aged particles. CN ($D_p > 0.01\ \mu m$) and UHSAS ($D_p > 0.07\ \mu m$) concentrations in the MBL ($CN_{MBL}$; $UHSAS_{MBL}$) and above the MBL inversion ($CN_{Inv}$; $UHSAS_{Inv}$) in Figure 7d and 8, respectively. To identify if MBL CN concentrations are higher, similar or lower to CN concentrations above the MBL inversion in Figure 7d, the vertical profiles of CN are divided into three subsections, corresponding to classification where $CN_{MBL}/CN_{Inv}$
$> 1.3$ (Figure 7a), $1.3 > CN_{MBL}/CN_{Inv} > 0.7$ (Figure 7b), and $CN_{MBL}/CN_{Inv} > 0.7$ (Figure 7c). Figures 7a-c show examples of two CN and $CCN_{0.43}$ vertical profiles. Figures 7a and 7c show profiles of CN concentrations when $CN_{MBL}/CN_{Inv} > 1$ (consistent with particle formation occurring in the MBL) and $CN_{MBL}/CN_{Inv} < 1$ (consistent with particle formation in the free troposphere or decoupled layer). When $CN_{MBL}/CN_{Inv} \sim 1$, particle formation has not recently occurred in either the MBL or above the inversion (Figure 7b), and mixing across the inversion homogenizes the aerosol concentrations between the free troposphere
and MBL. During this study, the $CN_{Inv}$ is generally greater $CN_{MBL}$, which suggests particle formation occurs more frequently above the MBL inversion, either in the free troposphere or a decoupled layer above the marine boundary layer. This is also shown in the histogram of the $CN_{MBL}/CN_{Inv}$ ratio (Figure 9a), which typically has a value of less than unity. These results are consistent with previous findings that the observed long-range transport of particles and their precursors from phytoplankton blooms (Figure 3d) typically occurs above the MBL (Hudson et al., 1998; Korhonen et al., 2008; Meskhidze and Nenes, 2006;
Russell et al., 1998; Sanchez et al., 2018; Thornton et al., 1997; Williamson et al., 2019; Yoon and Brimblecombe, 2002).

Similarly, Figure 8 compares UHSAS concentrations ($D_p > 0.07\ \mu m$) in the MBL to those above the MBL inversion. As the UHSAS provided vertical profiles of the aerosol, we use the UHSAS to complement the static CCN measurements to assess the vertical extent of cloud-active aerosol. $CCN_{0.3}$ and $CCN_{0.4}$ correlate well with UHSAS ($D_p > 0.07\ \mu m$) concentrations (r = 0.94). Contrary to the vertical extent of CN, UHSAS ($D_p > 0.07\ \mu m$) and $CCN_{0.43}$ concentrations are generally greater in the
MBL compared to above the MBL inversion (Figure 7a-c, Figure 9b), which suggests high MBL UHSAS concentrations resulted from the growth of Aitken mode particles to CCN sizes through cloud processing (Section 3.2.2) (Hudson et al., 1998) or gas-to-particle phase condensation in the MBL (Pirjola et al., 2004; Russell et al., 2007; Sanchez et al., 2018), and consequently associated with the Aged regime (Figure 8). Precipitation also likely played a role in depleting UHSAS and CCN-sized particles ($D_p > 0.07\ \mu m$) for the Scavenged regimes.

**4 Conclusions**





GV HIAPER airborne measurements collected during the Southern Ocean Clouds, Radiation Aerosol Transport Experimental Study (SOCRATES) of CN and CCN over the Southern Ocean (SO) during the austral summer were separated into four regimes based on back trajectories and CN-to-CCN ratios. Airborne CCN measurements were also compared to shipborne measurements on the *R/V Investigator* collected on the second Clouds, Aerosols, Precipitation, Radiation and atmospheric Composition Over the southeRn Ocean (CAPRICORN-2) campaign. The airborne measurements on the GV HIAPER show a weak gradient in CCN at 0.3% supersaturation ($CCN_{0.3}$) with increasing CCN concentrations to the south between 44°S to 62.1°S, which may be caused by aerosol precursors from Antarctic coastal biological emissions. Shipborne CCN measurements on the *R/V Investigator* also show gradients between 44°S to 67°S with a minimum around 60°S where the peak frequency of SO storm tracks occurs (Patoux et al., 2009). Enhanced ship-based CCN concentrations north of 50°S are likely from Australia, enhanced biogenic activity near the Australian coast, or even long-range transport from Antarctic coastal emissions. Elevated CCN concentrations to the south of 60°S originate from biogenic emissions from the Antarctic coastal area. The differences in the observed trends between airborne and shipborne CCN concentrations is likely due to the different sampling strategies.

The particle regimes from the GV measurements were determined from the observed bimodal distributions in CN and $CCN_{0.3}$ concentrations, with minimum values at 750 cm$^{-3}$ and 125 cm$^{-3}$, respectively. These minima were used as thresholds to identify different particle regimes. $CCN_{0.3}$ was used for this analysis because concentrations at 0.3% supersaturation showed the highest correlation with observed cloud droplet number concentrations (CDNCs). Four regimes have been identified based on back trajectories and CN and $CCN_{0.3}$ concentrations, which ranged from 116-1153 cm$^{-3}$ and 17-264 cm$^{-3}$, respectively. These regimes are labelled (1) Scavenged regime, with low CN and $CCN_{0.3}$ concentrations, (2) Scavenged + recent particle formation (RPF) regime, with high CN and low $CCN_{0.3}$ concentrations, (3) Aged regime, with low CN and high $CCN_{0.3}$ concentrations, and (4) Aged +RPF regime, with high CN and $CCN_{0.3}$ concentrations. Back trajectories associated with the Aged regime consistently intersected the Antarctic coast, an area with elevated phytoplankton biomass relative to the open ocean and a source of biogenic emissions. The Antarctic coastal emissions generate a latitudinal gradient in the UHSAS ($D_P > 0.07$ µm) and CCN concentrations, as well as a gradient in particle composition (inferred from hygroscopicity). The hygroscopicity gradient was derived from aerosol size distributions (UHSAS) and CCN spectra and resulted in less hygroscopic aerosol (lower κ) to the south, indicating CCN contained more biogenic sulfate and organics, relative to those further north, which likely contained a larger fraction of more hygroscopic sea salt. Biogenic emissions from coastal Antarctic areas accounted for most of the CCN and CDNC concentrations in the SO during the austral summer, while PMA only accounted for about 20% of observed CCN and CDNC.

Precipitation over the ~1.5-day trajectory inversely correlates with CCN concentrations, indicating precipitation scavenging is a major sink of CCN in the SO. The boundary layer cloud fraction was highest for the aged (high CCN) regime, suggesting cloud processing significantly enhanced CCN concentrations ($CCN_{0.3} = 185\pm38$ cm$^{-3}$ for the Aged regime) in non-precipitating clouds. High CN concentrations ($D_p > 0.01$ µm), characteristic of recent particle formation (RPF) corresponded to cases with low cloud fractions, which is consistent with particle formation in cumulus outflow, also found in previous studies (Bates et



al., 1998b; Clarke et al., 1999; Cotton et al., 1995; Perry and Hobbs, 1994). RPF is the main eventual source of CCN number
concentration in the SO. In addition, CN concentrations were typically lower in the MBL relative to concentrations above the
MBL, suggesting that RPF typically occurred above the MBL inversion – either in the free troposphere or a decoupled layer.
In contrast, CCN and particle concentrations with $D_p > 0.07$ µm (UHSAS) were higher in the MBL, suggesting growth of
recently formed particles to CCN sizes occurred after mixing into the MBL and subsequent aging through gas-to-particle
conversion and cloud processing.

Due to the remoteness of the SO, biogenic Antarctic coastal emissions appear to be the main CCN source to the SO MBL.
Long-range transport of these emissions is shown to enhance measured particle concentrations up to 2000 km away and
contribute significantly to the concentration and variability of SO CCN and CDNC. These results indicate that changes in
future coastal Antarctica SO phytoplankton production caused by climate change (Deppeler and Davidson, 2017) could have
significant ramifications for CCN concentrations and cloud properties in the SO. This work provides measurements that are
rare for this region of the globe, and may help reduce discrepancies between models and observations of CN and CCN
concentrations.

## Acknowledgements

K. J. Sanchez and G. C. Roberts acknowledge the support of NSF Grant No. AGS-1660374. G. Saliba and L. M. Russell
acknowledge NSF AGS-1660509. SOCRATES data are available through the following EOL UCAR repository
https://data.eol.ucar.edu/master_lists/generated/socrates/ (Sanchez and Roberts, 2018; UCAR/NCAR-Earth Observing
Laboratory, 2019). GDAS data are available at https://ready.arl.noaa.gov/HYSPLIT.php. Directions to obtain the ERA5 data
are available at https://confluence.ecmwf.int/display/CKB/How+to+download+ERA5#HowtodownloadERA5-1-
Introduction. Contains modified Copernicus Climate Change Service Information 2019. The Authors wish to thank the CSIRO
Marine National Facility (MNF) for its support in the form of sea time on RV Investigator, support personnel, scientific
equipment and data management. All data and samples acquired on the voyage are made publicly available in accordance with
MNF Policy. Processed *R/V Investigator* CCN data for CAPRICORN-2 are available at
https://www.cmar.csiro.au/data/trawler/survey_details.cfm?survey=IN2018_V01. Raw data is available by contacting the data
librarians (DataLibrariansOAMNF@csiro.au). We thank the UCAR/NCAR-Earth Observing Laboratory and the flight crew
for all the work done to obtain the measurements used in this manuscript. The Authors have no conflicts of interest to disclose.

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





**Table 1. Mean and standard deviation of CN and CCN$_{0.3}$ number concentration for the four identified regimes measured in the GV MBL legs.**

| Regime | CN (cm$^{-3}$) | CCN$_{0.3}$ (cm$^{-3}$) |
|---|---|---|
| Aged | 485±81 | 187±37 |
| Aged + RPF | 958±92 | 175±84 |
| Scavenged | 407±147 | 83±31 |
| Scavenged + RPF | 860±98 | 84±33 |






**Figure 1. Histograms and kernel density estimates of (a) CN concentrations and (c) MBL CCN$_{0.3}$ (CCN concentration at 0.3% supersaturation). (b) MBL CN and CCN$_{0.3}$. Measurements are divided into four particle regimes based on the observed bimodal distributions of both CN and CCN$_{0.3}$.**






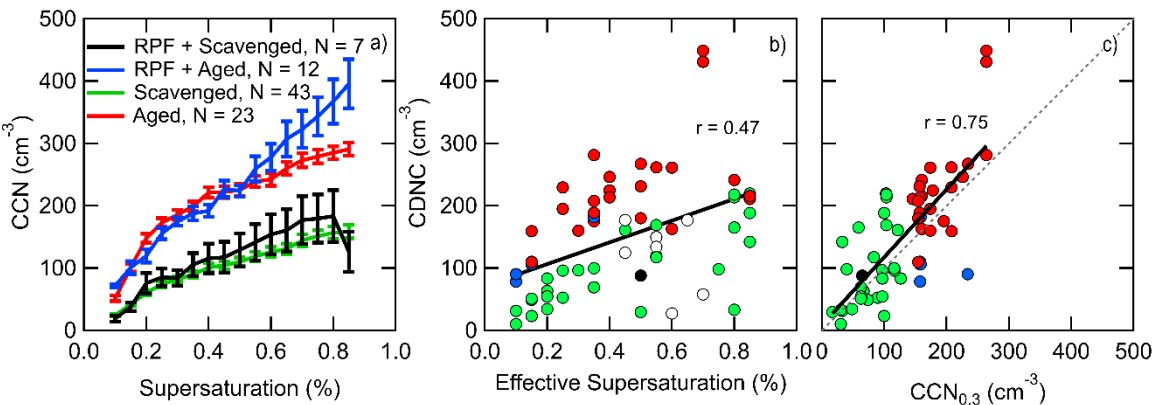

**Figure 2. (a) Mean MBL CCN spectra for each regime. The number of samples (N) at each supersaturation of the CCN spectra varied from the number of samples in the legend because occasionally CCN spectra scans were not fully completed by the end of the MBL leg. Error bars represent the standard error ($\sigma/\sqrt{N}$). Correlations of measured CDNC with (b) calculated effective supersaturation and (c) measured MBL CCN$_{0.3}$. Empty points did not have a corresponding CCN$_{0.3}$ or CN measurement. Solid lines in (b) and (c) represent linear fits and the dashed line in (c) represents the 1:1 line.**





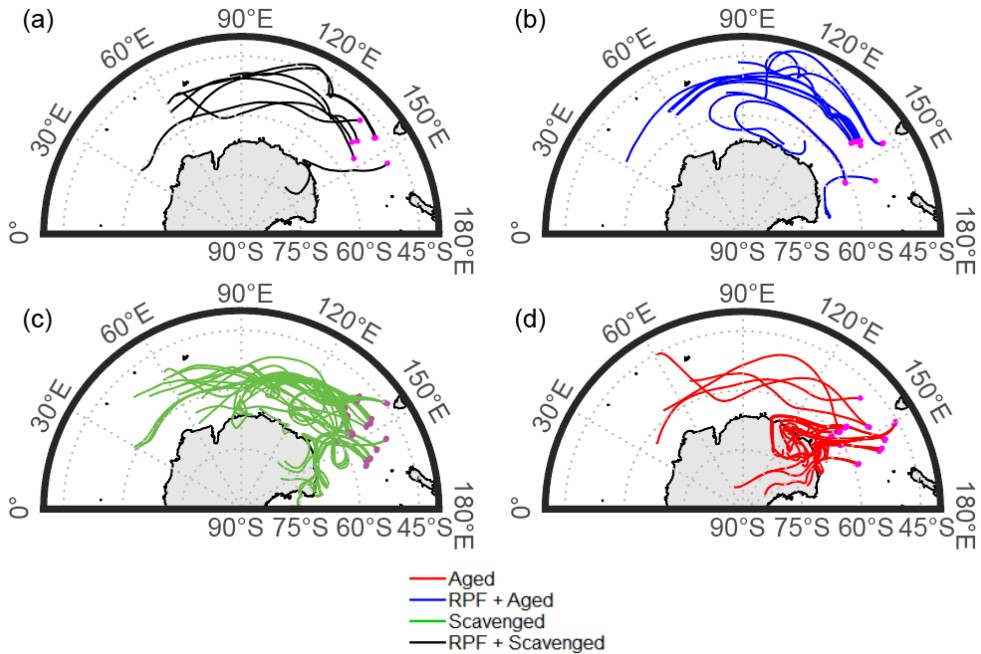

**Figure 3. Five-day HYSPLIT back trajectories starting from MBL legs (at 50-500 m amsl., magenta points) for each particle regime.**






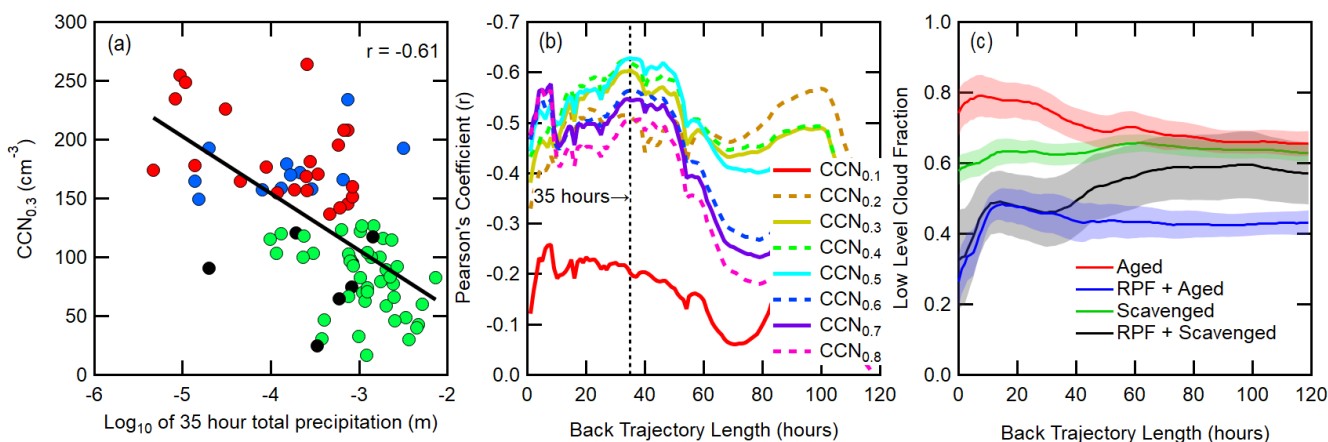

**Figure 4. (a) Correlation of MBL CCN$_{0.3}$ and total precipitation that occurred along a 35-hour HYSPLIT back trajectory. Marker colors correspond to legend in (c). (b) CCN and back trajectory total precipitation correlation coefficient as a function of back trajectory length. Vertical dashed line indicates peak in correlation with CCN$_{0.3}$ at 35-hours. (c) Particle regime averaged ERA5**

**low-level cloud fraction over the five-day back trajectory. Shaded areas represent one standard error.**


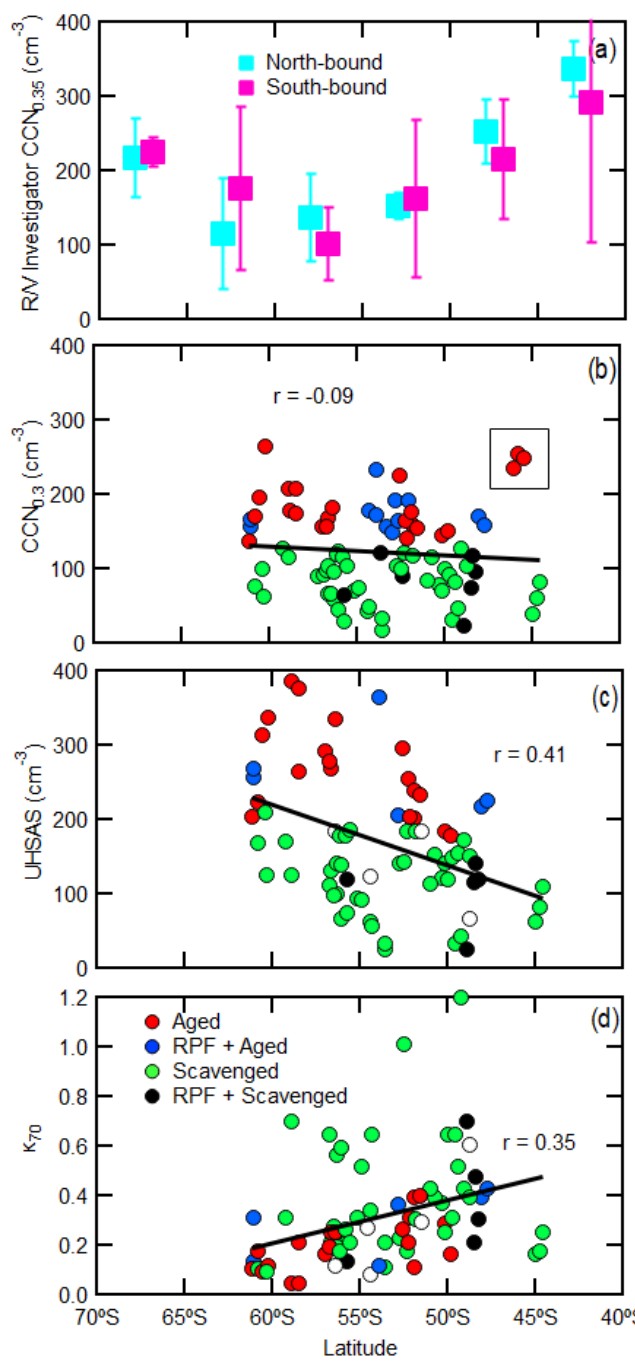

**Figure 5. (a) 5° latitude bin averaged CCN$_{0.35}$ from the *R/V Investigator*. Correlation of latitude to (b) HIAPER GV CCN$_{0.3}$ (c) total particle concentration with D$_p$ > 0.07 µm (UHSAS) and (d) κ derived at 0.07 µm. White points in (c) and (d) did not have a corresponding CCN$_{0.3}$ or CN measurement. Pearson's coefficient for (b) is r = -0.27 when excluding the three outliers at ~46°S highlighted in the black square.**





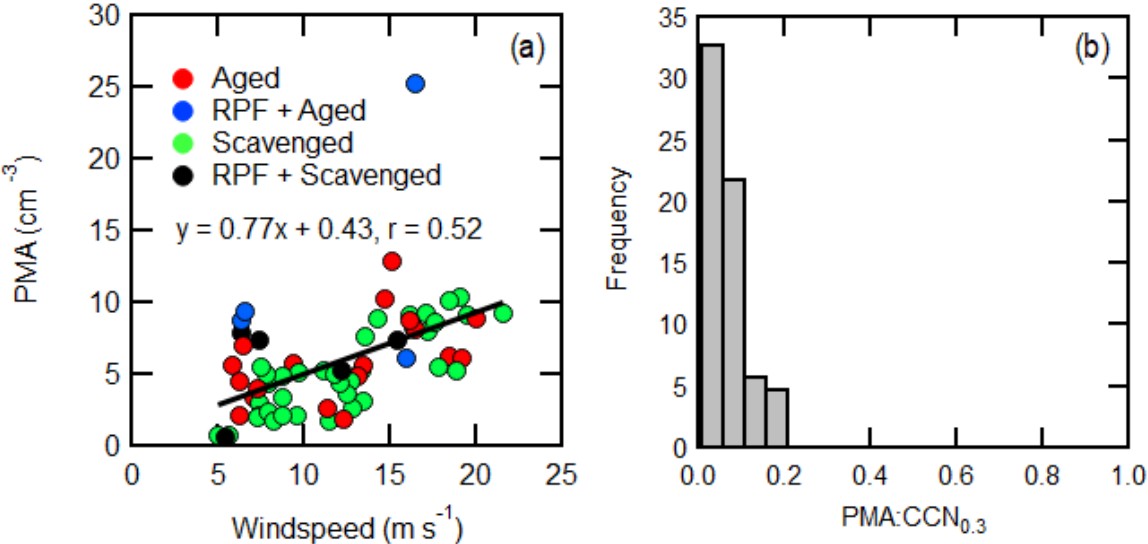

**Figure 6. (a) Correlation of estimated PMA concentration and wind speed, and (b) the fraction of PMA accounting for CCN$_{0.3}$ for MBL legs. Exclusion of the outlier in (a) increases the Pearsons Coefficient to 0.59 .**







**Figure 7. Vertical profiles of CN and CCN at 0.43% supersaturation corresponding to (a) elevated CN concentrations in the MBL, (b) well-mixed CN profiles and (c) elevated CN concentrations aloft. The cyan and magenta points in (a-c) represent 2 different vertical profiles. (d) Comparison of CN measured in the surface coupled MBL and decoupled layer or FT. Error bars represent standard error. Empty markers do not have a corresponding CCN$_{0.3}$ measurement.**



**Figure 8. Comparison of UHSAS concentrations ($D_P$ > 0.07 μm) measured in the surface coupled MBL and decoupled layer or FT. Error bars represent standard error. Empty markers do not have a corresponding $CCN_{0.3}$ measurement.**





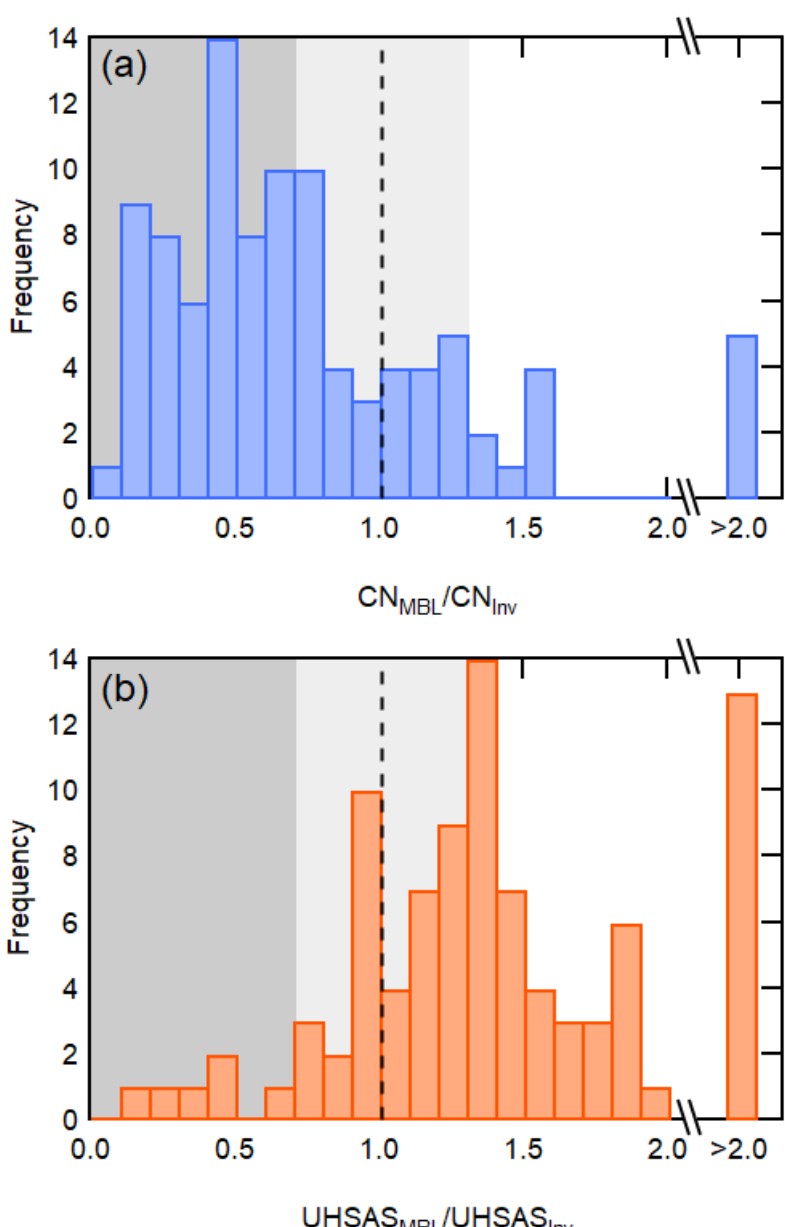

**Figure 9. Histogram of (a) CN_MBL/CN_Inv from Figure 7 and (b) UHSAS_MBL/UHSAS_Inv from Figure 8.**
