# Peer review of "Measurement Report: Cloud Processes and the Transport of Biological Emissions Affect Southern Ocean Particle and Cloud Condensation Nuclei Concentrations"

_Atmospheric Chemistry and Physics, 2020_

## Referee Comment (RC1) · Anonymous Referee #3 · 9 Oct 2020

The manuscript by Sanchez et al. "Measurement report: Cloud processes and the transport of biological emissions regulate Southern Ocean particle and cloud condensation nuclei concentration" reports on particle and cloud condensation nuclei measurements around the Southern Ocean during the SOCRATES and the CAPRICORN-2 campaigns. The measurements from SOCRATES were interpreted as a combination of high and low CN and CCN concentrations (four combinations) and linked to back trajectories and fitted with a course-mode fitting procedure to isolate the PMA contribution to CCN and CDNC. In parallel CCN measurements were conducted on-board a

research vessel (during CAPRICORN-2). Overall the measurements are very interesting and novel and should be published as a measurement report. I do however find that the manuscript needs improvement in several aspects. In particular I miss information about measurement methods.

**General comments**

**1. Title**: I find that the title should be revised. I think that the use of the word *regulates* is too strong and not justified.

**2. CCN measurements**

**(a)** Is the miniature CCNc (mCCNc) used and described in more details in literature? Or is it a miniature custom-made version of the one described in Roberts and Nenes, 2005?

**(b)** Details on the supersaturation calibration procedures is missing (both the mCCNc and DMT CCNc). Could the authors elaborate on this (in the manuscript or in data repository) and give details on how they estimate the instruments supersaturation? (e.g. assumed water activity? type of calibration aerosol? drying conditions? uncertainty related to reported supersaturation?). I find it valuable and useful that the data is available online. Nice that you estimate the uncertainty in the CCN number concentration.

**(c)** What caused the scanning mode mCCNc instrument to have problems during RF01-05? How is this reflected in the uncertainty?

**(d)** How dry was the air before entering the CCN counters?

**3. Hygroscopicity** Another major concern is related to the $\kappa$ parameter and hygroscopicity. Could the authors elaborate on how the $\kappa_{70}$ was derived from scanning CCN spectra and UHSAS number size distributions? Did the authors estimate an $\kappa$ uncertainty?

**Specific comments**

L63: Could iodine new particle formation be a source in the SO? e.g. see the recent work of Baccarini et al. (2020) and Sipila et al. (2016).

L101-103: "some studies suggest biologically productive waters enhance PMA production". Some studies show this relationship is not that simple. I suggest modifying the text to also reflect on this. See e.g. the work of Collins et al. (2016), Bates et al. (2020)

L108-110: "... can further reduce the hygroscopicity". A couple of studies also show that the hygroscopicity is not changing that much. I suggest the text to also reflect on this, e.g. see the recent work by Bates et al. (2020), Christiansen et al. (2020).

L170-172: Some details on aerosol inlet, sampling, calibration, data analysis or a reference to a paper/data repository giving these details is missing.

L193: "UHSAS particles in the SOCRATES campaign were not fully dried". Could the author explain how this would affect the derived hygroscopicity parameter $\kappa$?

L200-204: A bit confused here: "The TEM analysis shows almost all particles > 0.2 $\mu$m in diameter consist of sea salt; and sea salt particles account for 25% to 100% of particle number concentrations at particle diameters > 0.4 $\mu$m (Saliba et al. submitted)." How can almost all > 0.2 $\mu$m be sea salt, but not particles > 0.4 $\mu$m? How much is *almost all*? What kind of diameter is this (geometric, optical, mobility)? In addition, the title of the **Saliba et al. submitted** states "...in the North Atlantic". Was

SOCRATES field campaign also in the North Atlantic?

L212: Kernel density estimates. Interesting approach. Did you try k-means clustering analysis? If yes, how do they compare?

L232: The **Aged regime**: in addition, could aging be caused by atmospheric oxidants (OH ect.), e.g. changing chemical composition due to chemical reactions?

L265: Sanchez et al., 2017. Still in discussion for ACP? Wrong citation.

L280: FT not defined I think.

L317: Would prefer to use **lower** $\kappa$ instead of smaller $\kappa$.

L319-320: "... relative to sea salt ($\kappa$ = 1.3)". It has become clear that the hygroscopicity of NaCl and sea salt are not the same (Zieger et al., 2017; Christiansen et al., 2020) and that the $\kappa$ value of sea salt $\sim$ 1 rather than 1.3. On a side note, Bates et al. (2020) writes about the CCN results in Quinn et al. (2014) that "It is critical to thoroughly dry the aerosol before it enters the CCN instrument. The SSA was not sufficiently dry during the WACS‐1 measurements".

L333: "...phytoplankton emissions...". To help the reader, please be more specific here on what type of emission.

L347-348: "The calculated PMA number concentrations moderately correlated to wind speed (r = 0.53, Figure 6a)". This is interesting. Why did you use a linear fit here?

L447-448: I can not find the processed CCN data for CAPRICORN-2 / *RV Investigator* through the provided link.

*References:*

Baccarini, A., Karlsson, L., Dommen, J. et al. Frequent new particle formation over the high Arctic pack ice by enhanced iodine emissions. Nat Commun 11, 4924 (2020). https://doi.org/10.1038/s41467-020-18551-0

Sipilä, M., Sarnela, N., Jokinen, T. et al. Molecular-scale evidence of aerosol particle formation via sequential addition of HIO3. Nature 537, 532–534 (2016). https://doi.org/10.1038/nature19314

Bates, T. S., Quinn, P. K., Coffman, D. J., Johnson, J. E., Upchurch, L., Saliba, G., . . . Behrenfeld, M. J. (2020). Variability in Marine Plankton Ecosystems Are Not Observed in Freshly Emitted Sea Spray Aerosol Over the North Atlantic Ocean. Geophysical Research Letters, 47(1). https://doi.org/10.1029/2019GL085938

Christiansen, S., Ickes, L., Bulatovic, I., Leck, C., Murray, B. J., Bertram, A. K., . . . Bilde, M. (2020). Influence of Arctic microlayers and algal cultures on sea spray hygroscopicity and the possible implications for mixed‐phase clouds. Journal of Geophysical Research: Atmospheres, 125(19). https://doi.org/10.1029/2020jd032808

Collins, D. B., Bertram, T. H., Sultana, C. M., Lee, C., Axson, J. L., Prather, K. A. (2016). Phytoplankton blooms weakly influence the cloud forming ability of sea spray aerosol. Geophysical Research Letters, 43(18), 9975–9983. https://doi.org/10.1002/2016GL069922

Zieger, P., Väisänen, O., Corbin, J. C., Partridge, D. G., Bastelberger, S., Mousavi-Fard, M., . . . Salter, M. E. (2017). Revising the hygroscopicity of inorganic sea salt particles. Nature Communications, 8, ncomms15883. https://doi.org/10.1038/ncomms15883

**Technical corrections**
Overall, there are quite many mistakes in the reference list. This gives the reader a bad impression. The authors should solve this issue.

L452: Remove uppercase "AEROSOLS, CLOUD MICROPHYSICS, AND FRACTIONAL CLOUDINESS"

L463: Replace + with page number 363. "359-+,"

L467: What journal? Only says "aerosols, , (November), 1–27". No DOI.

L470: Remove uppercase "SEASONAL RELATIONSHIP BETWEEN CLOUD CONDENSATION NUCLEI AND AEROSOL METHANESULFONATE IN MARINE AIR"

L495: Fix "67-+,"

L517: Remove uppercase "NEW PARTICLE FORMATION IN THE MARINE BOUNDARY-LAYER,"

L634: Is the * supposed to be there? "...Low Cloud Feedback*"

[Figure]

L670: ?? "Available from: %3CGo,"

L674: Make lowercase

L681: Make lowercase

L701: Remove symbol

L735: Who is "O'". Maybe cite the final paper.

L742: Make lowercase

L757: No title??? What journal?

L801: Make lowercase

L808: Make lowercase

Table 1: $\pm$ what?

[Figure]

---

## Referee Comment (RC2) · Luke Cravigan (Referee) · 9 Oct 2020

General comments Sanchez et al. (Cloud Processes and the Transport of Biological Emissions Regulate Southern Ocean Particle and Cloud Condensation Nuclei Concentrations) provides a very good summary of airborne and marine boundary layer aerosol and CCN measurements over the Southern Ocean. These are extremely valuable measurements and provide excellent information on aerosol sources, atmospheric processing and the resulting CCN/CDNC. I have outlined some minor changes to the manuscript. A little more detail on the calculation and interpretation of the hygroscop-

icity, and on the uncertainties would improve the clarity of the manuscript. Specific comments are outlined below.

Specific comments

L 41: What are the conclusions regarding the latitudinal gradient in particle composition and hygroscopicity based on? Are there measurements for this?

L 41: Isn't there biogenic/coastal emissions in the lower latitude Southern Ocean, and therefore lower sea salt fraction?

L 54: Which particle sources?

L 84: Alroe et al. doesn't show such dramatic increase in number concentrations, and was a summertime voyage.

L 92: This paragraph could be simplified. The discussion of PMA (e.g. from "In a recent study.." onward) could be separated out from the discussion of seasonality.

L 93: Biological particles is an unclear term, perhaps something like secondary particles from biological sources. Biogenic particles is used on L95.

L123: The sentence beginning "In contrast, the inclusion..." would be better placed in a paragraph about PMA.

L125: The sentence beginning "SO satellite-derived....", begins the discussion of cloud properties and could be the start of a new paragraph.

L120: Which sensitivity? McCoy et al. doesn't discuss aerosol growth via cloud processing.

L149: Do we have any idea to what RH the sample was dried?

L159: Some detail on how kappa was calculated is required. I assume the supersaturation at which the CCNc = UHSAS # conc (>0.07um) was taken as the Scrit. Then the Scrit/Dcrit were used to compute the kappa. Although I'm not sure how it was computed, I don't it was measured at 0.07 um (i.e. pre-selected). Therefore, perhaps the expression "the 0.07 $\mu$m diameter hygroscopicity parameter ($\kappa$70)" is a bit misleading.

L195: How do the authors know that the fitted PMA distribution isn't a subset of the total PMA? Particularly the deliquesced mode at ~0.6 um, since the GSD ~1.44 is on the lower end of that observed by Saliba et al. 2019. Does the sensitivity testing that was done in Saliba et al. 2019 necessarily transfer across to these data?

L199: What are the mode (and sd) for the fitted PMA distribution? and how do they vary? Only N is reported here.

L208/Fig 1b: What is the uncertainty/variability in these averaged CN/CCN? Error bars would be helpful.

L228: "Aitken-mode particles (CN)" is unclear and suggests CN excludes the accumulation mode.

L249: "(RPF + Aged and RPF + Scavenged)"

L256: What was the rationale for taking the 90th percentile? and how wasa this implemented? the 90th percentile for each vertical profile?

L256/Fig 3c: It would be good to represent the variability CDNC, and how this varibility influences the relationship between CCN and CDNC.

Fig 3: Error bars are required in Figure 3b and c.

L266: The sentence starting "The two regimes.." should be revised. The figure shows that the CDNC for aged is greater than CDNC for scavenged, there is too little data to draw conclusions for the RPF cases.

L310: Does removing these transported outliers result in statistical significance in the trend ? to what p-value?

Fig 5: Could error bars be added to the CCN, UHSAS and kappa?

[Figure]

L315: It is not clear how the CCN and CN (>0.07 um) concentrations by themselves imply anything about composition/hygroscopicity.

L319: The computed kappa values are lower than that expected for sulfates, particularly at high SO latitudes. At lower latitudes the kappa values are consistent with sulfates. This should be noted/discussed.

L332: Seawater biogenics (e.g. chl-a) increase in the low latitude SO, north of the sub-antarctic front. This isn't necessarily associated with the Aus continent. McCoy et al. 2015 pointed out that secondary aerosols played a more important role in CCN/CDNC in the lower latitude SO, with the contribution from PMA increasing with latitude up to 60S.

L346: In general the variability using this method is quite high. How does the variability compare with that in Saliba et al.

L414/L41: The hygroscopicity values are low and aren't necessarily consistent with sea salt in the north and sulfates (+organics) in the south.

Technical questions

L29: "coarse-mode" not "course-mode"

L101: "though" is not necessary

L235: RPF regimes exhibit

L370: I think it should be "< 0.7" not "> 0.7"

---

## Referee Comment (RC3) · Anonymous Referee #1 · 21 Oct 2020

The manuscript analyzes measurements of particle number concentrations (CN) and cloud condensation nuclei (CCN) obtained by airborne measurements during SOCRATES and by ship-borne observations during CAPRICORN-2 in the Australian sector of the Southern Ocean. The study comprehensively shows the effects of cloud processing, precipitation and air mass origin on particle size and number, and on CCN. To this end the authors combine direct observations and re-analyses data. They also show nicely that in most cases new particles are formed in the free troposphere and not in the marine boundary layer.. These measurements make an important contribution

to our understanding of CN and CCN processes in the Southern Ocean. I recommend that the paper be published, however, only after considering below points, which I believe will help improve the study. Also, if possible, I recommend that this manuscript be published as a normal research article rather than a measurement report. The depth of analysis is not untypical of that in research articles.

General comments:

A map with the cruise and flights tracks is needed.

The discussion of NPF in the Southern Ocean (SO) boundary layer is partly incorrect, because it is said that the condensation sink (CS) is low. See specific comments in the attachment and also further below.

Some statements about the Southern Ocean are too general, e.g., the claim that microbial activity is low compared to the Antarctic coastal region. There are hotspots, like South Georgia, and if measurements had been taken in that region of the Southern Ocean the paper would report different observations. Hence acknowledging the regional variability of the SO is very important. Otherwise incorrect messages about this large region are published. This comment is also true for the conclusions. See comments in the attached.

The introduction jumps between topics, particularly ll. 92 – 110. The main message is not clear. I suggest to structure this part of the introduction as follows: observations of NPF, CN and CCN near the Antarctic coast, observations of NPF, CN and CCN over the open southern ocean (and not only between Australia and Antarctica), then discuss how the coastal and open ocean regions are connected, then go deeper into cloud processing.

Some more details in the methodology section are needed, in particular regarding the mini CCNCs and the calculation of kappa. Also the inlet system and position of the CCNC on R/V Investigator is not described. In addition, it is unclear why the CCN data

were not compared at the same supersaturation. It is possible to interpolate from the spectrum. See attachment for more specific comments.

The calculation of back trajectories is not well described. If it is really the case that only one location per leg was used, the results will be highly uncertain. Some more clarification is needed, particularly a better description of flight legs.

Specific comments:

l. 193: A quantification or at least better approximation of the RH in the sample flow is needed to make this study comparable to previous and future studies.

l. 202: the description of the fraction of PMA to particles $> 0.2$ $\mu$m is inconsistent (see attached comments).

l. 229: An explanation of how the Aitken mode was derived is missing. The UHSAS was only used for particles greater 70 nm, so cannot have been used for that purpose. Was the CPC data used? If yes, how were CPC and UHSAS intercompared?

Section 3.3 on cloud processing relies strongly on ERA 5 data. Some discussion on the representation of clouds and particularly precipitation in the reanalysis product is needed. Over the SO there are not many observations that would constrain the reanalysis.

Section 3.4 Latitudinal Gradient: Recent observations by Schmale et al. (2019) also highlight the higher concentrations of CCN near the Antarctic coast. See also their discussion of kappa for MSA and the role of particle size to activate as CCN. I recommend referring to their work in section 3.4, since they already came to similar conclusions presented in section 3.4.

l. 346: Do the authors means the low variability +/- 0.04? Why would a wet diameter lead to a lower variability?

In section 3.5 PMA Marine Aerosol, again it would be useful to put the results into

context with recent publication from other sectors of the SO. Schmale et al. (2019) show in their table 3 the contribution of their similarly identified sea spray mode to CCN and find between 20 and 30 % for SS = 0.15 %. l. 363 The low condensation sink (CS) is not really true because the presence of sea spray leads to such a high condensation sink that new particle formation in the marine boundary layer is rather an exception (also due to other factors). Compared to the Arctic Ocean (Baccarini et al. (2020), https://doi.org/10.1038/s41467-020-18551-0), the CS in the SO will be a factor four, or even more, higher. The authors have the necessary data to actually calculate the CS. Compared to other oceans (except polar oceans) the CS might be lower, but given the low new particle formation occurrence, saying low CS is not completely correct.

l. 366, which trend?

Section 3.6, please provide the number of data points per vertical profile. It is difficult to understand how representative the six profiles from figure 7 are and why particularly those were chosen. How many were there? How was the histogram in Fig. 9 calculated, is there one ratio per profile? L. 400: The explanation of long-range transported CCN from the Antarctic coast is in contradiction to the minimum near 60°S. If the higher concentrations near the coast of Australia are due to specific long-range transport events, this should be said explicitly.

l. 392 f: The information on the four regimes is repeated in l. 404ff. Consider removing some redundancy from the conclusions.

Please also note the supplement to this comment:
https://acp.copernicus.org/preprints/acp-2020-731/acp-2020-731-RC3-supplement.pdf

———————————————————

[Figure]

**Supplement:**

[revised manuscript text omitted]

---

## Author Comment (AC1) · 19 Dec 2020

The manuscript by Sanchez et al. "Measurement report: Cloud processes and the transport of biological emissions regulate Southern Ocean particle and cloud condensation nuclei concentrations" reports on particle and cloud condensation nuclei measurements around the Southern Ocean during the SOCRATES and the CAPRICORN- 2 campaigns. The measurements from SOCRATES were interpreted as a combination of high and low CN and CCN concentrations (four combinations) and linked to back trajectories and fitted with a course-mode fitting procedure to isolate the PMA contribution to CCN and CDNC. In parallel CCN measurements were conducted on-board a research vessel (during CAPRICORN-2). Overall the measurements are very interesting and novel and should be published as a measurement report. I do however find that the manuscript needs improvement in several aspects. In particular I miss information about measurement methods.

**General comments**
**1. Title**: I find that the title should be revised. I think that the use of the word regulates is too strong and not justified.

We have changed 'regulates' to 'affect'.

**2. CCN measurements**
**(a)** Is the miniature CCNc (mCCNc) used and described in more details in literature? Or is it a miniature custom-made version of the one described in Roberts and Nenes,
2005?
The minature CCNc is a custom-made version described by Roberts and Nenes 2005.

**(b)** Details on the supersaturation calibration procedures is missing (both the mCCNc and DMT CCNc). Could the authors elaborate on this (in the manuscript or in data repository) and give details on how they estimate the instruments supersaturation? (e.g. assumed water activity? type of calibration aerosol? drying conditions? Uncertainty related to reported supersaturation?). I find it valuable and useful that the data is available online. Nice that you estimate the uncertainty in the CCN number concentration.
The procedure for calibrating the miniature CCN counter is identical to the method described by Roberts and Nenes 2005. In summary, empirical calibrations are derived using monodisperse ammonium sulfate particles that are dried and then measured by the CCN counter and a CN counter to derive the activated fraction. Kohler theory is used to derive the supersaturation (assuming a water activity of 1.0). An instrument model, discussed in Roberts and Nenes (2005), showed a standard deviation in the supersaturation estimate of about +/-0.01%.

In the manuscript we have added the following statement to address the questions on CCN details:
"The miniature CCN counters are custom-made and operate with the same physical principles as described by Roberts and Nenes (2005). Empirical calibrations are derived using dried monodisperse ammonium sulfate particles that are measured by the CCN counter and a CN counter to derive the activated fraction. Kohler theory is used to derive the supersaturation (assuming a water activity of 1.0). An instrument model, discussed in Roberts and Nenes (2005) showed a standard deviation in the supersaturation estimate of about +/-0.01%."

**(c)** What caused the scanning mode mCCNc instrument to have problems during RF01-05? How is this reflected in the uncertainty?

During the first five research flights, the scanning CCN counter laser optical threshold was mistakenly set too high. Typically, the CCN counter is set to count particles that grow to over 1 micrometer in diameter. With the optical threshold elevated, the counting threshold was greater than 1 micrometer, causing the instrument to only count a fraction of the activated CCN, particularly at the lower supersaturations. Post-experiment lab calibrations corrected these measurements by applying a scaling factor between 1.0 to 1.6 as a function of supersaturation. The observed standard deviation ($\sigma$) in the scaling factor was less than 7% for all supersaturations and was included in the standard error ($\sigma/\sqrt{N}$) calculation.

**(d)** How dry was the air before entering the CCN counters?

The air was dried before entering the CCN counter by heating in the inlet lines inside the GV cabin; however, that should not have any effect on the number of particles activated. Note that CCN calibrations are done using dry air (~ 40%). Anything more humid will not affect the results, and only drier air at the operating limits of the CCN (lowest supersaturation) could have a measurable impact.

**3. Hygroscopicity** Another major concern is related to the kappa parameter and hygroscopicity. Could the authors elaborate on how the $\_{70}$ was derived from scanning CCN spectra and UHSAS number size distributions? Did the authors estimate an kappa$_{70}$ uncertainty?

The text has been updated to better describe Kappa70:

> "The CCN spectra and UHSAS number concentrations on the GV were used to estimate the hygroscopicity parameter at 0.07 μm diameter ($\kappa_{70}$) for each MBL leg. For this calculation, the critical supersaturation is derived from the CCN spectra, where the UHSAS concentration at 0.07 μm diameter is equivalent to the CCN concentration."

The uncertainty (standard deviation) in the kappa value has been estimated and is now in figure 6d shown below.

[Figure]

L63: Could iodine new particle formation be a source in the SO? e.g. see the recent work of Baccarini et al. (2020) and Sipila et al. (2016).

Baccarini associates the NPF with increasing iodine during the transition from Arctic summer to autumn when the Arctic Ocean refreezes or ozone levels rise. Neither one of these conditions were observed during SOCRATES experiments. However, we cannot rule out the contribution of iodine emissions as a driver of NPF, particularly in the Antarctic coastal waters. No changes have been made to the text.

L101-103: "some studies suggest biologically productive waters enhance PMA production". Some studies show this relationship is not that simple. I suggest modifying the text to also reflect on this. See e.g. the work of Collins et al. (2016), Bates et al. (2020)

This sentence has been updated:

"Some studies suggest biologically productive waters enhance PMA production (Fuentes et al, 2010), while others suggest that biogenic content has little to no influence on PMA production (Collins et al., 2016; Bates et al., 2020). "

L108-110: "... can further reduce the hygroscopicity". A couple of studies also show that the hygroscopicity is not changing that much. I suggest the text to also reflect on this, e.g. see the recent work by Bates et al. (2020), Christiansen et al. (2020).
We have added the suggested citations to the discussion and have changed "can" to "may".

L170-172: Some details on aerosol inlet, sampling, calibration, data analysis or a reference to a paper/data repository giving these details is missing.

The following text has been added:
> "Details of the aerosol sampling system on board the RV Investigator are presented in Humphries et al. (2019) and Alroe et al., (2020). In short, aerosol sampling occurred via a common sampling inlet mounted on a mast at the bow of the ship, located 18 m above sea level. The CCN counter sampled from a manifold located 8 m below the mast in the ship's bow."

> "The full CCN data set collected on the RV Investigator during CAPRICORN2 are available at Humphries et al., (2020)."

L193: "UHSAS particles in the SOCRATES campaign were not fully dried". Could the author explain how this would affect the derived hygroscopicity parameter _?
After further discussion with instrument operators, we have concluded that the RH is expected to be reduced to less than 40% based on results from Strapp et al. (1992). Strapp et al. (1992) showed that the PCASP-100X dries aerosol almost completely (<40% RH) with the de-icing heaters. The UHSAS and PCASP are made by the same company (DMT) and have identical de-icing front ends. On a GV, there is additional heating related to ram air, consequently, we consider that the UHSAS air as completely dried (also < 40% RH). Despite this finding, we believe the PMA mode did not completely dry because of the presence of the 600 nm mode that correlates with wind speed. We hypothesize that the low residence time (~0.2 seconds) of the aerosol in the instrument prevented the larger hygroscopic sea salt from fully drying before being measured. This mode is only found in the MBL and not the FT, further suggesting its presence is a result of a deliquesced PMA mode. The deliquesced sea salt particles are not expected to affect the total UHSAS particle concentration, and therefore not affect the derived kappa parameter using the UHSAS lower cut off diameter of 70 nm.

We have updated the text to state

"PMA particles in the SOCRATES campaign were not fully dried"

We also added the following text:

"This deliquesced mode was present despite the findings by Strapp et al. (1992), suggesting the de-icing heaters of the PCASP-100X (which are identical to those used for the UHSAS) is expected to dry the particles to less than 40% relative humidity. We hypothesize that the low residence time of the aerosol in the instrument (~0.2 seconds) prevented the large hygroscopic sea salt from fully drying before being measured."

L200-204: A bit confused here: "The TEM analysis shows almost all particles >0.2 _m in diameter consist of sea salt; and sea salt particles account for 25% to 100% of particle number concentrations at particle diameters > 0.4 _m (Saliba et al. submitted)." How can almost all > 0.2 _m be sea salt, but not particles > 0.4 _m? How much is almost all? What kind of diameter is this (geometric, optical, mobility)? In addition, the title of the **Saliba et al. submitted** states "...in the North Atlantic". Was SOCRATES field campaign also in the North Atlantic?

This statement has been simplified to improve clarity:

"The TEM analysis showed that ~70-95% of marine boundary layer particles > 0.5 um optical diameter are PMA sea spray (Twohy et al. submitted)."

The previous version of this statement reference the wrong manuscript and should have referenced the manuscript listed below, which was a part of SOCRATES. However, the reference was changed to the more appropriate Twohy et al. submitted, which focused on the TEM analysis.

G. Saliba, K. J. Sanchez, L. M. Russell, C.H. Twohy, G.C. Roberts, S. Lewis, J. Dedrick, C.S. McCluskey, K. Moore, P.J. DeMott & D.W. Toohey (2020) Organic Composition of Three Different Size Ranges of Aerosol Particles over the Southern Ocean, Aerosol Science and Technology, DOI: 10.1080/02786826.2020.1845296

L212: Kernel density estimates. Interesting approach. Did you try k-means clustering analysis? If yes, how do they compare?

Yes, we tried a k-means clustering approach and the cases were categorized very similarly, and there were only a few differences at the thresholds between categories. We elected to use the kernel density approach as the visual showing the bimodality of the CN and UHSAS concentrations is more intuitive than the k-means clustering approach.

L232: The **Aged regime**: in addition, could aging be caused by atmospheric oxidants (OH ect.), e.g. changing chemical composition due to chemical reactions?

We absolutely believe aging is caused by atmospheric chemical reactions driven by oxidants. The text in the manuscript implies that ageing can also be caused by atmospheric oxidants:

The aged regime is "...likely due to atmospheric processes that increase particle size over time such as the condensation of VOC oxidation products or cloud processing."

L265: Sanchez et al., 2017. Still in discussion for ACP? Wrong citation.

The citation has been fixed.

L280: FT not defined I think.

We have now defined FT as free troposphere

L317: Would prefer to use **lower** _ instead of smaller _.
We agree and have changed "smaller kappa" to "lower kappa".

L319-320: "... relative to sea salt (_ = 1.3)". It has become clear that the hygroscopicity of NaCl and sea salt are not the same (Zieger et al., 2017; Christiansen et al., 2020) and that the _ value of sea salt _ 1 rather than 1.3. On a side note, Bates et al. (2020) writes about the CCN results in Quinn et al. (2014) that "It is critical to thoroughly dry the aerosol before it enters the CCN instrument. The SSA was not sufficiently dry during the WACSăˇARˇ 1 measurements".
We agree that a hygroscopicity value of 1 is more representative of primary marine aerosol and have modified the text to indicate a value of 1.0 and cited Zieger et al. 2017 and Christiansen et al. 2020.

Regarding the comment about drying the air before it enters the CCN instrument, it is important to have dry classified aerosol (and correct for shape) when calibrating the CCN instruments. In Quinn et al. (2014), instrument setup was classifying the non-dry (deliquesced) SSA before sampling by the CCN and CN, which leads to incorrect calculation of solute mass for a given activation diameter – this has nothing to do with the operation of the CCN instrument. Once a CCN instrument is properly calibrated, one does not need to dry the aerosol. Caveat: The only condition where a measurable impact could occur is when operating the CCN instrument close to its low supersaturation limit in an environment significantly dryer than it was calibrated. In such a case, there is a small delay in the establishment of the peak supersaturation inside the column, which can reduce the residence time of droplet growth and lead to the undercounting of the activated aerosol particles.

L333: "...phytoplankton emissions...". To help the reader, please be more specific here on what type of emission.
We have modified the text based on the comments from reviewer 2 and 3:
  "There are also elevated CCN concentrations north of 50°S measured on the R/V Investigator, probably related to continental emissions from Australia, elevated biomass emissions of VOCs (aerosol precursors), as suggested by increasing chlorophyll-a concentrations north of the subantarctic front (McCoy et al. 2015), and even long-range transport of Antarctic coastal emissions (Ayers and Gillett, 2000; Twohy et al., submitted)."

L347-348: "The calculated PMA number concentrations moderately correlated to wind speed (r = 0.53, Figure 6a)". This is interesting. Why did you use a linear fit here?

The relationship between PMA and wind speed is non-linear (i.e., Grythe et al. ACP, 2014), and we have changed the linear fit to a polynomial fit. The r value remains 0.53.

[Figure]

L447-448: I can not find the processed CCN data for CAPRICORN-2 / RV Investigator through the provided link.

The link has been updated and works.

References:
Baccarini, A., Karlsson, L., Dommen, J. et al. Frequent new particle formation over the high Arctic pack ice by enhanced iodine emissions. Nat Commun 11, 4924 (2020). https://doi.org/10.1038/s41467-020-18551-0
Sipilä, M., Sarnela, N., Jokinen, T. et al. Molecular-scale evidence of aerosol particle formation via sequential addition of HIO3. Nature 537, 532–534 (2016). https://doi.org/10.1038/nature19314
Bates, T. S., Quinn, P. K., Coffman, D. J., Johnson, J. E., Upchurch, L., Saliba, G., : : : Behrenfeld, M. J. (2020). Variability in Marine Plankton Ecosystems Are Not Observed in Freshly Emitted Sea Spray Aerosol Over the North Atlantic Ocean. Geophysical Research Letters, 47(1).
https://doi.org/10.1029/2019GL085938 Christiansen, S., Ickes, L., Bulatovic, I., Leck, C., Murray, B. J., Bertram, A. K., : : : Bilde, M. (2020). Influence of Arctic microlayers and algal cultures on sea spray hygroscopicity and the possible implications for mixedˇARˇ phase clouds. Journal of Geophysical Research: Atmospheres, 125(19). https://doi.org/10.1029/2020jd032808
Collins, D. B., Bertram, T. H., Sultana, C. M., Lee, C., Axson, J. L., Prather, K. A. (2016). Phytoplankton blooms weakly influence the cloud forming ability of sea spray aerosol. Geophysical Research Letters, 43(18), 9975–9983. https://doi.org/10.1002/2016GL069922
Zieger, P., Väisänen, O., Corbin, J. C., Partridge, D. G., Bastelberger, S., Mousavi-Fard, M., : : : Salter, M. E. (2017). Revising the hygroscopicity of inorganic sea salt particles. Nature Communications, 8, ncomms15883. https://doi.org/10.1038/ncomms15883

**Technical corrections**
Overall, there are quite many mistakes in the reference list. This gives the reader a bad impression. The authors should solve this issue.
There were some errors when importing documents into the reference manager that were not changed before submission. We thank the reviewer for taking the time to point these out.
L452: Remove uppercase "AEROSOLS, CLOUD MICROPHYSICS, AND FRACTIONAL CLOUDINESS"

L463: Replace + with page number 363. "359-+,"
L467: What journal? Only says "aerosols, , (November), 1–27". No DOI.
L470: Remove uppercase "SEASONAL RELATIONSHIP BETWEEN CLOUD CONDENSATION NUCLEI AND AEROSOL METHANESULFONATE IN MARINE AIR"
L495: Fix "67-+,"
L517: Remove uppercase "NEW PARTICLE FORMATION IN THE MARINE BOUNDARY-LAYER,"
L634: Is the * supposed to be there? "...Low Cloud Feedback*"
L670: ?? "Available from: %3CGo,"
L674: Make lowercase
L681: Make lowercase
L701: Remove symbol
L735: Who is "O'". Maybe cite the final paper.
L742: Make lowercase
L757: No title??? What journal?
L801: Make lowercase
L808: Make lowercase

Table 1: _ what?
The authors do not understand this last comment.

---

## Author Comment (AC2) · 19 Dec 2020

General comments Sanchez et al. (Cloud Processes and the Transport of Biological Emissions Regulate Southern Ocean Particle and Cloud Condensation Nuclei Concentrations) provides a very good summary of airborne and marine boundary layer aerosol and CCN measurements over the Southern Ocean. These are extremely valuable measurements and provide excellent information on aerosol sources, atmospheric processing and the resulting CCN/CDNC. I have outlined some minor changes to the manuscript. A little more detail on the calculation and interpretation of the hygroscopicity, and on the uncertainties would improve the clarity of the manuscript. Specific comments are outlined below.

Specific comments
L 41: What are the conclusions regarding the latitudinal gradient in particle composition and hygroscopicity based on? Are there measurements for this?
The text has been updated to indicate the measurement hygroscopicity is based on combining CCN spectra and particle size distributions. See updated text in response to the next comment.
L 41: Isn't there biogenic/coastal emissions in the lower latitude Southern Ocean, and therefore lower sea salt fraction?
Yes, there are biogenic/coastal emissions near the Antarctic coast that are transported northward, but the main reason for the lower sea salt fraction in these lower latitudes is a result of less wind and the presence of sea ice (consistent with findings from Schmale et al. (2019)).
We believe the reviewer is referring to the Australian coastal/biogenic emissions. Almost all of our measurements are at latitudes higher than 50S, putting them well away from the Australian coastal influence. Furthermore, the back trajectories are all from the west/south and therefore, are not heavily influenced by the Australian coast. We have added that the gradient is derived from measurements south of the Australian coast:

"In addition, a latitudinal gradient in the particle composition, south of the Australian and Tasmanian coast, is apparent in aerosol hygroscopicity derived from CCN spectra and aerosol particle size distribution. The particles are more hygroscopic to the north, consistent with a greater fraction of sea salt from PMA, and less hygroscopic to the south as there is more sulfate and organic particles originating from biogenic sources in coastal Antarctica."

L 54: Which particle sources?
We have changed "these particle sources" to "Southern Ocean particle sources".
L 84: Alroe et al. doesn't show such dramatic increase in number concentrations, and was a summertime voyage.
We agree, Alroe et al. showed a less dramatic increase. We have discovered that the Alroe et al. manuscript has been updated to its final published form, which no longer contains plots showing the particle concentration as a function of latitude. For this reason, we have removed the reference to Alroe et al. here, but left the reference in other sentences where it is still appropriate. The following reference has also been added to the results section:

"The storm track frequency peaks around 60°S (Patoux et al., 2009), suggesting parcels of air entering the storm track from the south have also been influenced by coastal Antarctic biogenic DMS and other VOC emissions, eventually leading to increases in CCN concentrations via cloud processing and in the absence of precipitation. Schmale et al, 2019 and Alroe et al. 2020 also

find that the higher fraction of particles serving as CCN near the coast of Antarctica, are also from biologically derived particles."

L 92: This paragraph could be simplified. The discussion of PMA (e.g. from "In a recent study.." onward) could be separated out from the discussion of seasonality.

This introduction has been reorganized based on suggestions by multiple reviewers:

[revised manuscript text omitted]

L 93: Biological particles is an unclear term, perhaps something like secondary particles from biological sources. Biogenic particles is used on L95.
"biological particles" has been changed to "secondary particles from biological emissions"

L123: The sentence beginning "In contrast, the inclusion..." would be better placed in a paragraph about PMA.
We agree and have moved this sentence to the paragraph related to PMA.

L125: The sentence beginning "SO satellite-derived....", begins the discussion of cloud properties and could be the start of a new paragraph.
We agree and have started a new paragraph here.

L120: Which sensitivity? McCoy et al. doesn't discuss aerosol growth via cloud processing.
McCoy et al. did not specifically discuss aerosol growth via cloud processing, but they did discuss the general complexity of aerosol processes that lead to natural aerosols influencing clouds, CDNC and overall radiative forcing.
We have updated the text as follows to prevent confusion.

> "In the event that the cloud droplets do not precipitate, the evaporated particles are larger than their original size since non-volatile compounds (i.e., sulfate and MSA) that condensed onto the cloud droplet remain in the particle phase. This added mass eventually shifts Aitken mode particles to the accumulation mode (Hoppel et al., 1986; Hudson et al., 2015; Kaufman and Tanré, 1994; Sanchez et al., 2017). Results from McCoy et al. (2015) emphasized that, despite the ambiguous results from focused modeling and observational studies of such aerosol processes, their general global model simulations of natural aerosol account for more than half the spatial and temporal variability in the satellite derived CDNC over the SO."

L149: Do we have any idea to what RH the sample was dried?
Strapp et al. (1992) showed that the PCASP-100X dries aerosol almost completely (<40% RH) with the deicing heaters. The UHSAS and PCASP are made by the same company (DMT), and have identical deicing front ends. On a GV, there is additional heating related to ram air, consequently, we consider that the UHSAS air as completely dried (also < 40% RH).
We added the following text:

> "This deliquesced mode was present despite the findings by Strapp et al. (1992), suggesting the de-icing heaters of the PCASP-100X (which are identical to those used for the UHSAS) is expected to dry the particles to less than 40% relative humidity. We hypothesize that the low

L159: Some detail on how kappa was calculated is required. I assume the supersaturation at which the CCNc = UHSAS # conc (>0.07um) was taken as the Scrit. Then the Scrit/Dcrit were used to compute the kappa. Although I'm not sure how it was computed, I don't it was measured at 0.07 um (i.e. pre-selected). Therefore, perhaps the expression "the 0.07 _m diameter hygroscopicity parameter (_70)" is a bit misleading.

The UHSAS distribution lower cut off size of 0.07 μm diameter particles was used as the critical diameter in the kappa derivation; hence, the 70 notation. We have updated the text to explain how the critical supersaturation and diameter were determined.

Updated text:

"The CCN spectra and UHSAS number concentrations on the GV were used to estimate the hygroscopicity parameter at 0.07 μm diameter ($\kappa_{70}$) for each MBL leg. For this calculation, the critical supersaturation is derived from the CCN spectra, where the UHSAS concentration at 0.07 μm diameter is equivalent to the CCN concentration."

L195: How do the authors know that the fitted PMA distribution isn't a subset of the total PMA? Particularly the deliquesced mode at _0.6 um, since the GSD _1.44 is on the lower end of that observed by Saliba et al. 2019. Does the sensitivity testing that was done in Saliba et al. 2019 necessarily transfer across to these data?

To test if the fitted PMA distribution is not only a subset of the total PMA distribution, we added various fractions of the remaining UHSAS distribution to see if the correlation with wind speed would improve (figure below), similar to the sensitivity test performed by Saliba et al. 2019. The correlation only decreased suggesting the fitted PMA distribution is representative of the total PMA.

[Figure]

Reviewer Figure 1

Since this deliquesced mode exists, it suggests the particles have not reached efflorescence, which is based on the relative humidity and particle composition independent of size for larger particles if at equilibrium (Cheng et al., 2015). However, if the particles are limited by mass or heat transfer rates and are not at equilibrium, then it may be that larger particles are slower to dry out than smaller particles. While typically observed in cloud droplet distributions, condensational growth leads to narrower and larger distributions (Pinsky et al., 2014; Yum and Hudson, 2005), so it is not surprising that the GSD is on the lower end of what was observed by Saliba et al. (2019).

Note the efflorescence RH of sea salt (~47%) is higher than the estimated drying RH (<40%) suggested by Strapp et al. (1992). We hypothesize that the low residence time in the PCASP (~0.2 seconds) was not enough for the larger hygroscopic sea salt to fully dry before being measured. In addition, the fact that we do not see this mode in the free tropospheric measurements supports that this is a PMA mode.

We added the following text to address this concern:

> "This deliquesced mode was present despite the findings by Strapp et al. (1992), suggesting the deicing heaters of the PCASP-100X (which are identical to those used for the UHSAS) is expected to dry the particles to less than 40% relative humidity. We hypothesize that the low residence time of the aerosol in the instrument (0.2 seconds) prevented the large hygroscopic sea salt from fully drying before being measured."

L199: What are the mode (and sd) for the fitted PMA distribution? and how do they vary? Only N is reported here.
The following sentence has been updated to include the mode diameter and the sd of the fitted PMA distributions.

> "The estimated PMA mode diameter and geometric width (0.59 ± 0.04 µm, 1.44±0.25, respectively) are consistent with sea salt distributions (from PMA) observed on size-resolved particles collected in the marine boundary layer during SOCRATES and analyzed with transmission electron microscopy (TEM)."

L208/Fig 1b: What is the uncertainty/variability in these averaged CN/CCN? Error bars would be helpful.
The figure has been updated to include error bars representing the standard error. The standard error in CN concentration is sufficiently small so that the error bars are within the size of the marker.

[Figure]

L228: "Aitken-mode particles (CN)" is unclear and suggests CN excludes the accumulation mode.
We agree and have changed the text to "Aitken + accumulation-mode particles (CN)"

L249: "(RPF + Aged and RPF + Scavenged)"
Fixed.

L256: What was the rationale for taking the 90th percentile? and how wasa this implemented? the 90th percentile for each vertical profile?
The 90th percentile was used to consistently exclude cloud droplet measurements that were heavily influenced by entrainment drying. The 100th percentile was not used to avoid outliers. The text has been modified and is shown below to clearly state our reasoning:
    "For this comparison, the 90th percentile of CDNC from each vertical profile is matched to the nearest below-cloud MBL leg CCN concentration. The use of the 90th percentile of CDNC excludes measurements that are heavily influenced by entrainment drying and also excludes outliers."

L256/Fig 3c: It would be good to represent the variability CDNC, and how this variability influences the relationship between CCN and CDNC.

We agree this would be ideal; however the CDNC were generally measured during vertical profiles and are susceptible to variations at cloud top/cloud base. Horizontal legs through cloud layers would be better for identifying the variability in CDNC, but there were few such legs during SOCRATES.

Fig 3: Error bars are required in Figure 3b and c.
Since we used the 90th percentile rather than an average (and since we measured CDNC during vertical profiles), we do not have a good representation of the CDNC horizontal variability. We have updated the figure to include the horizontal variability of CCN during low level legs.

[Figure]

L266: The sentence starting "The two regimes.." should be revised. The figure shows that the CDNC for aged is greater than CDNC for scavenged, there is too little data to draw conclusions for the RPF cases.
We agree and have revised the sentence to the following:
> "The two regimes with aged particles (high CCN) consistently had higher CDNCs than the scavenged regime, highlighting the role of CCN concentrations as CDNC."

L310: Does removing these transported outliers result in statistical significance in the trend ? to what p-value?
Removing the outliers results in a weak correlation (r = -0.26; p-value < 0.05).

Fig 5: Could error bars be added to the CCN, UHSAS and kappa?
1-sigma error bars have been added to the figure and is shown below:

[Figure]

L315: It is not clear how the CCN and CN (>0.07 um) concentrations by themselves imply anything about composition/hygroscopicity.

As mentioned earlier in the text, the CCN spectra and UHSAS number concentrations were used to estimate the hygroscopicity parameter at ($\kappa_{70}$) by relating the UHSAS concentration at 0.07 µm to a CCN concentration at a given supersaturation. The lower cut size of the UHSAS (0.07 µm diameter) serves as the critical diameter in determining kappa. We have updated the text to better explain how the data indicates there is a north-south variation in particle composition:

"A latitudinal gradient is observed in both the UHSAS particle (Dp > 0.07 µm) and CCN concentrations; however, the differences in their slopes imply a north-south gradient in particle

composition (i.e., hygroscopicity) across the SO, as identified by the hygroscopicity parameter (κ) for Dp > 0.07 μm (Figure 6d)."

L319: The computed kappa values are lower than that expected for sulfates, particularly at high SO latitudes. At lower latitudes the kappa values are consistent with sulfates. This should be noted/discussed.
The sentence has been revised:
"The lower κ (less hygroscopic aerosol) at high latitudes is consistent with sulfates and organic aerosol from biogenic emissions, which have relatively low κ values (κ = 0.6-0.9 and κ < 0.2, respectively) compared to PMA (κ ~ 1.0 (Quinn et al., 2014)) present in primary emissions at the lower latitudes."

L332: Seawater biogenics (e.g. chl-a) increase in the low latitude SO, north of the subantarct ic front. This isn't necessarily associated with the Aus continent. McCoy et al. 2015 pointed out that secondary aerosols played a more important role in CCN/CDNC in the lower latitude SO, with the contribution from PMA increasing with latitude up to 60S.
We agree with this statement have updated the text and cited McCoy et al. 2015:
"There are also elevated CCN concentrations north of 50°S measured on the R/V Investigator, probably related to continental emissions from Australia, elevated biomass emissions of VOCs (aerosol precursors), as suggested by increasing chlorophyll-a concentrations north of the subantarctic front (McCoy et al. 2015), and even long-range transport of Antarctic coastal emissions (Ayers and Gillett, 2000; Twohy et al., submitted)."

L346: In general the variability using this method is quite high. How does the variability compare with that in Saliba et al.

The variability in the mode diameter is actually low compared to Saliba et al. 2019. Please see updated text:
"The mode diameter of the retrieved PMA number size distribution was 0.59 ± 0.04 μm, which is consistent with the average mode diameter observed in the North Atlantic of 0.54 ± 0.21 μm (Saliba et al., 2019). The low geometric width (1.44±0.25) of the PMA mode, relative to Saliba et al. (2019) (ranging from 1.5-4.0), likely reflects the available statistics (N = 74), and the possibility that the PMA particles were not completely dry (section 2.3)."
L414/L41: The hygroscopicity values are low and aren't necessarily consistent with sea salt in the north and sulfates (+organics) in the south.
We agree that the hygroscopicity values are not "consistent" with sea salt but are "consistent with a greater fraction of sea salt".

Technical questions
L29: "coarse-mode" not "course-mode" Fixed.
L101: "though" is not necessary Fixed.
L235: RPF regimes exhibit Fixed.
L370: I think it should be "< 0.7" not "> 0.7" Fixed.

---

## Author Comment (AC3) · 19 Dec 2020

The manuscript analyzes measurements of particle number concentrations (CN) and cloud condensation nuclei (CCN) obtained by airborne measurements during SOCRATES and by ship-borne observations during CAPRICORN-2 in the Australian sector of the Southern Ocean. The study comprehensively shows the effects of cloud processing, precipitation and air mass origin on particle size and number, and on CCN. To this end the authors combine direct observations and re-analyses data. They also show nicely that in most cases new particles are formed in the free troposphere and not in the marine boundary layer. These measurements make an important contribution to our understanding of CN and CCN processes in the Southern Ocean. I recommend that the paper be published, however, only after considering below points, which I believe will help improve the study. Also, if possible, I recommend that this manuscript be published as a normal research article rather than a measurement report. The depth of analysis is not untypical of that in research articles.

We thank the reviewer for the detailed and thoughtful comments, which have improved the overall quality of the manuscript. For convenience, specific comments that the reviewer included in the supplement document have all been copied and responded to at the end of this review. The article was initially submitted with the intent of a normal research article; however, at the editor's request, we resubmitted the article as a measurement report. Based on the reviews, the decision of how to publish will be left for the editor.

General comments:
A map with the cruise and flights tracks is needed.

We have added the following figure with the cruise and flight tracks.

[Figure]

Figure 1. SOCRATES and CAPRICORN-2 study region. Blue and red lines represent the SOCRATES flight tracks and CAPRICORN-2 *R/V Investigator* tracks, respectively.

The discussion of NPF in the Southern Ocean (SO) boundary layer is partly incorrect, because it is said that the condensation sink (CS) is low. See specific comments in the attachment and also further below. We have addressed all of the comments on condensation sink in detail in the responses below.

Some statements about the Southern Ocean are too general, e.g., the claim that microbial activity is low compared to the Antarctic coastal region. There are hotspots, like South Georgia, and if measurements had been taken in that region of the Southern Ocean the paper would report different observations. Hence acknowledging the regional variability of the SO is very important. Otherwise incorrect messages about this large region are published. This comment is also true for the conclusions. See comments in the attached.

We acknowledge that there are hotspots and have addressed all comments about the region of the Southern Ocean SOCRATES and CAPRICORN-2 measurements below.

The introduction jumps between topics, particularly ll. 92 – 110. The main message is not clear. I suggest to structure this part of the introduction as follows: observations of NPF, CN and CCN near the Antarctic coast, observations of NPF, CN and CCN over the open southern ocean (and not only between Australia and Antarctica), then discuss how the coastal and open ocean regions are connected, then go deeper into cloud processing.
This introduction has been reorganized based on suggestions by multiple reviewers:
   "The remote mid-latitude SO contains much less biological activity near the ocean surface relative to the Antarctic continental coast, which creates a latitudinal gradient in the contribution of particles from biogenic sources, with the exception of some biological hotspots such as near South Georgia (Alroe et al., 2020; Humphries et al., 2016; Kim et al., 2019; O'Dowd et al., 1997; O'Shea et al., 2017; Schmale et al., 2019; Weller et al., 2018). Shipborne observations in the region south of Australia show a distinct increase in aerosol concentrations south of 64°S, where CN concentrations are about five times higher during the austral spring months (Humphries et al., 2016). The seasonal variability of biogenically derived particles is linked to seasonal variations in SO biological activity (Ayers and Gras, 1991; Korhonen et al., 2008). On the Antarctic peninsula, NPF events occurred mostly during the austral summer, with CCN concentrations (at 0.4% supersaturation) increasing on average by 11% (Kim et al., 2019). Similarly, higher average concentrations of cloud droplet number concentrations (CDNC) are observed in the austral summer (Mace and Avey, 2017; McCoy et al., 2015). Some studies suggest biologically productive waters enhance PMA production (Fuentes et al., 2010), while other studies find that biogenic content has little to no influence on PMA production (Bates et al., 2020; Collins et al., 2016). In any case, PMA CCN is found to have little seasonal variability relative to biogenic CCN (Vallina et al., 2006), likely driven by small seasonal differences in wind speed (Saliba et al., 2019). Organic enrichment of PMA in biologically productive waters may further reduce their hygroscopicity (Burrows et al., 2018; Cravigan et al., 2019; Law et al., 2017; Meskhidze and Nenes, 2010).

   Long-range transport of aerosol and gaseous precursors in the MBL and free troposphere from the Antarctic continental coast plays a significant role in increasing CN, CCN, and CDNC concentrations in the SO (Bates et al., 1998a; Clarke et al., 1998, 2013; Dzepina et al., 2015; Korhonen et al., 2008; Woodhouse et al., 2010). With substantial growth of newly formed particles by the uptake of VOC oxidation products through cloud processing, particles from

biogenic sources may grow CCN larger and subsequently increase CDNC (Hoppel et al., 1986; Hudson et al., 2015; Pirjola et al., 2004; Russell et al., 2007; Sanchez et al., 2018). Cloud processing occurs when small particles activate to form cloud droplets, leading to enhanced condensation of VOC oxidation products onto the droplet because the droplet surface area is larger than that of the unactivated particles. Aqueous phase oxidation of absorbed VOCs also results in the formation of less volatile compounds, which remain in the particle phase upon evaporation of the water (Hoppel et al., 1986). In the event that the cloud droplets do not precipitate, the evaporated particles are larger than their original size since aqueous oxidation of volatile compounds (i.e., DMS, MSA, $SO_2$ and nitric acid) have formed non-volatile sulfates and nitrates that remain in the particle phase. This added mass eventually shifts Aitken mode particles to the accumulation mode (Hoppel et al., 1986; Hudson et al., 2015; Kaufman and Tanré, 1994; Sanchez et al., 2017; Schmale et al., 2019). Results from McCoy et al. (2015) show that, despite the ambiguous results from focused modelling and observational studies of such aerosol processes, their general global model simulations of natural aerosol account for more than half the spatial and temporal variability in the satellite-derived CDNC over the SO. These areas of enhanced CDNC also correlate with areas of high chlorophyll-a, a tracer for phytoplankton activity, which increases secondary sulfate and organic aerosol concentrations (Krüger and Grabßl, 2011; McCoy et al., 2015). SO satellite-derived cloud properties such as liquid water content (LWC), effective radius, and cloud fraction showed seasonal variations that resulted in a difference in cloud radiative forcing (i.e., surface cooling) between 14 and 23 W m-2 (Mccoy et al., 2014). Increased CDNC is also shown to correlate with enhanced cloud fraction, significantly increasing overall cloud shortwave forcing (Rosenfeld et al., 2019). If cloud droplets precipitate, CN and CCN concentrations are reduced through precipitation scavenging (Croft et al., 2010; Stevens and Feingold, 2009)."

Some more details in the methodology section are needed, in particular regarding the mini CCNCs and the calculation of kappa. Also the inlet system and position of the CCNC on R/V Investigator is not described. In addition, it is unclear why the CCN data were not compared at the same supersaturation. It is possible to interpolate from the spectrum. See attachment for more specific comments.
We have added the following text to describe the calculation of kappa:
"The CCN spectra and UHSAS number concentrations on the GV were used to estimate the hygroscopicity parameter at 0.07 μm diameter ($κ_{70}$) for each MBL leg. For this calculation, the critical supersaturation is derived from the CCN spectra, where the UHSAS concentration at 0.07 μm diameter is equivalent to the CCN concentration."
We have also added the following text to clearly identify details of the mini CCNc can be found:
"The miniature CCN counters are custom-made and operate with the same physical principles described by Roberts and Nenes (2005)."
The following text was added to address inlet system on the R/V Investigator.
"Details of the aerosol sampling system on board the RV Investigator are presented in Humphries et al. (2019) and Alroe et al., (2020). In short, aerosol sampling occurred via a common sampling inlet mounted on a mast at the bow of the ship, located 18 m above sea level. The CCN counter sampled from a manifold located 8 m below the mast in the ship's bow."

We have updated Figure 5a (now figure 6a) to show the R/V Investigator CCN concentration at 0.30% SS by interpolating the concentrations measured at 0.25% and 0.35% (shown below).

[Figure]

The calculation of back trajectories is not well described. If it is really the case that only one location per leg was used, the results will be highly uncertain. Some more clarification is needed, particularly a better description of flight legs.

We have updated the text as follows:

> "The latitude, longitude and altitude (50-500 m) averaged for each CCN spectra (~150 seconds, ~15-20 km horizontal distance) collected during the MBL legs on the GV HIAPER were used as starting points for the back trajectories."

Specific comments:

l. 193: A quantification or at least better approximation of the RH in the sample flow is needed to make this study comparable to previous and future studies.

Strapp et al. 1992 showed that the PCASP-100X dries aerosol almost completely (<40% RH) with the deicing heaters. The UHSAS and PCASP are made by the same company (DMT) and have identical deicing front ends. On a GV, there is additional heating related to ram air, consequently, we consider that the UHSAS air as completely dried (also <40% RH).

We added the following text:

> "This deliquesced mode was present despite the findings by Strapp et al. (1992), suggesting the de-icing heaters of the PCASP-100X (which are identical to those used for the UHSAS) is expected to dry the particles to less than 40% relative humidity. We hypothesize that the low residence time of the aerosol in the instrument (~0.2 seconds) prevented the large hygroscopic sea salt from fully drying before being measured."

l. 202: the description of the fraction of PMA to particles > 0.2 _m is inconsistent (see attached comments).

This statement has been simplified to improve clarity:

> "The TEM analysis showed that ~70-95% of marine boundary layer particles > 0.5 um optical diameter are PMA sea spray (Twohy et al., submitted)."

l. 229: An explanation of how the Aitken mode was derived is missing. The UHSAS was only used for particles greater 70 nm, so cannot have been used for that purpose. Was the CPC data used? If yes, how were CPC and UHSAS intercompared?

The Aitken mode was not directly derived, and the following text has been modified to prevent confusion:

> "The classification of each regime is based on the relative concentration of Aitken + accumulation-mode particles (CN) and accumulation-mode particles (CCN sizes), with a naming convention that describes the corresponding airmass history. Similar to analyses in previous studies, the relative contribution of the accumulation-mode to the total particle concentration is used to identify recent particle formation (RPF) events and growth of small (<70 nm diameter) particles to accumulation-mode or CCN sizes (Kalivitis et al., 2015; Kleinman et al., 2012; Williamson et al., 2019). The Scavenged regime is named based on evidence indicating the removal of CCN-sized particles through precipitation scavenging (Section 3.3). The Aged regime represents cases in which accumulation-mode is prominent and CCN particle concentrations are relatively high, likely due to atmospheric processes that increase particle size over time such as the condensation of VOC oxidation products or cloud processing (Section 3.2 and 3.3, respectively). The RPF regimes exhibit a high CN concentration (>10 nm diameter), indicative of recent particle formation (Section 3.2)."

Section 3.3 on cloud processing relies strongly on ERA 5 data. Some discussion on the representation of clouds and particularly precipitation in the reanalysis product is needed. Over the SO there are not many observations that would constrain the reanalysis.

We agree that discussion of the ERA5 data set is needed, specifically on the lack of observations and have added the following text to section 3.3:

"Manton et al. (2020) showed the ERA5 annual cycle of precipitation across the SO is consistent with in-situ data, but it is important to note that there is large uncertainty because of the low number of observations to constrain the ERA5."

"Similar to ERA5 precipitation, there are also a low number of observations to constrain the ERA5 cloud fraction product. Ship measurements in the region south of Australia were recently shown to be consistent with daily averaged observations and ERA5 cloud fraction values of 0.75±0.23 and 0.71±0.27, respectively, providing some confidence in the ERA5 (Wang et al. 2020)."

Section 3.4 Latitudinal Gradient: Recent observations by Schmale et al. (2019) also highlight the higher concentrations of CCN near the Antarctic coast. See also their discussion of kappa for MSA and the role of particle size to activate as CCN. I recommend referring to their work in section 3.4, since they already came to similar conclusions presented in section 3.4.

We have modified this section based on comments from all the reviewers. Below is the relevant updated text:

"A latitudinal gradient is observed in both the GV HIAPER UHSAS particle (Dp > 0.07 µm) and CCN concentrations; however, the differences in their slopes imply a north-south gradient in particle composition (i.e., hygroscopicity) across the SO, as identified by the hygroscopicity parameter ($\kappa_{70}$) for Dp > 0.07 µm (Figure 6d). The presence of a latitudinal gradient in aerosol concentrations (Dp > 0.07 µm) and a weak gradient in the GV HIAPER CCN implies a north-south gradient in particle composition (i.e., hygroscopicity) across the SO. Figure 6d shows the hygroscopicity parameter ($\kappa$) for Dp > 0.07 µm derived at each MBL leg. The lower $\kappa$ (less hygroscopic aerosol) at high latitudes is consistent with sulfates and organic aerosol from biogenic emissions, which have relatively low $\kappa$ values ($\kappa$ = 0.6-0.9 and $\kappa$ < 0.2, respectively) compared to PMA ($\kappa \sim 1.0$ (Quinn et al., 2014)) present in primary emissions at the lower latitudes (Kreidenweis and Asa-Awuku, 2013; Petters and Kreidenweis, 2007). These results are consistent with findings of Schmale et al. (2019) showing MSA, an aerosol component associated with biogenic emissions, contributed about 2.5 times more mass in the Antarctic coastal region compared to the remote SO. Furthermore, the elevated CCN near the Antarctic coast is also consistent with a higher incidence of cloud processing in the region, despite the lower particle hygroscopicity (Alroe et al., 2020; Schmale et al., 2019)."

l. 346: Do the authors means the low variability +/- 0.04? Why would a wet diameter lead to a lower variability?

We have updated the text to more clearly indicate that a wet diameter could lead to a low geometric width. While typically observed in cloud droplet distributions, condensational growth leads to narrower distributions (Pinsky et al., 2014; Yum and Hudson, 2005).

"The mode diameter of the retrieved PMA number size distribution was 0.59 ± 0.04 µm, which is consistent with the average mode diameter observed in the North Atlantic of 0.54 ± 0.21 µm (Saliba et al., 2019). The low geometric width (1.44±0.25) of the PMA mode, relative to Saliba et al., (2019) (ranging from 1.5-4.0) likely reflects the available statistics (N = 74), and the possibility that the PMA particles were not completely dry (section 2.3)."

In section 3.5 PMA Marine Aerosol, again it would be useful to put the results into context with recent publication from other sectors of the SO. Schmale et al. (2019) show in their table 3 the contribution of their similarly identified sea spray mode to CCN and find between 20 and 30 % for SS = 0.15 %.

We have added this relevant finding in the following text:

"These results are consistent with Twohy et al. (submitted) who found sea-spray aerosol comprised a minority of cloud droplet residual number in three SOCRATES cases. Similarly, Quinn et al. (2017) who found that PMA contributed to less than 30% of CCN number concentrations (at 0.3% supersaturation) from measurements collected during other field campaigns conducted between 130°E (near Tasmania) and eastward to 60°W (near South America) . Also, Schmale et al. (2020) showed that over three measurement legs that spanned the entire longitudinal range of the SO, the average PMA contribution to CCN ranged from 19-32% at a supersaturation of 0.15%."

l. 363 The low condensation sink (CS) is not really true because the presence of sea spray leads to such a high condensation sink that new particle formation in the marine boundary layer is rather an exception (also due to other factors). Compared to the Arctic Ocean (Baccarini et al. (2020), https://doi.org/10.1038/s41467-020-18551-0), the CS in the SO will be a factor four, or even more, higher. The authors have the necessary data to actually calculate the CS. Compared to other oceans (except polar oceans) the CS might be lower, but given the low new particle formation occurrence, saying low CS is not completely correct.

We agree that the presence of sea spray leads to a high condensation sink. However, we are pointing out that new particle formation often happens in the FT and not the MBL, even in the SO, despite the remoteness of the SO to continental and anthropogenic influences. In a recently published article (Sanchez et al. 2020; https://doi.org/10.5194/acp-2020-702) this exact calculation was performed using sea spray concentrations in the North Atlantic to show that under high wind conditions (> 10 m s$^{-1}$), the sea spray particles account for a significant fraction of the total particle surface area, and less so under low wind speed conditions. It is unclear as to why the reviewer states that the CS in the SO will be a factor of four or more higher compared to the arctic. There seems to be no mention of the SO in Baccarini et al. 2020, and little on sea spray specifically as a CS.

To better describe the CS in the SO we have updated the text as followed:
"During this study, the CN$_{Inv}$ is generally greater than CN$_{MBL}$, which suggests particle formation occurs more frequently above the MBL inversion, either in the free troposphere or a decoupled layer above the marine boundary layer. Despite the lack of influence from continental and anthropogenic particles as condensational sinks in the SO, the presence of a small concentration of PMA particles can lead to a high total particle surface area (Sanchez et al. 2020; Cainey and Harvey, 2002; Yoon and Brimblecombe, 2002) and prevent new particle formation in the MBL."

l. 366, which trend?
We have updated the statement as follows:
"To determine if the SO MBL truly is an exception to the trend of NPF typically occurring in the FT, we compare the concentrations of FT and BL CN and UHSAS concentrations across the MBL."

Section 3.6, please provide the number of data points per vertical profile. It is difficult to understand how representative the six profiles from figure 7 are and why particularly those were chosen. How many were there? How was the histogram in Fig. 9 calculated, is there one ratio per profile?

Each data point in Figure 7 (now figure 8) represents a 4 ± 2 minute vertical profile of measurements made at 1 Hz. Consequently, the number of data points per vertical profile differ depending on the range of the vertical profile and presence of a cloud as shown in figure 7a-c (now figure 8a-c). Showing the vertical profiles are not necessary to make the point we have made about comparisons between CN concentration in the MBL and above inversion layers. The six profiles are provided as examples from each section of Figure 7d (now figure 8d). Overall, there were 110 profiles with measurements in both layers (MBL and above the inversion). The histograms in Figure 8 and 9 (now figure 9 and 10) are derived directly from the points in Figure 7 and Figure 8 (now figure 8 and 9) as indicated in the Figure 9 (now figure 10) description (one ratio per profile).

L. 400: The explanation of long-range transported CCN from the Antarctic coast is in contradiction to the minimum near 60_S. If the higher concentrations near the coast of Australia are due to specific long-range transport events, this should be said explicitly.

The text has been updated to indicate that a specific case of higher concentrations near Australia is found to be due to long-range transport from near the coast of Australia.

> "Enhanced ship-based CCN concentrations north of 50°S are likely from Australia. In one case enhanced CCN concentration measured on the GV near the Australian coast is shown to be from long-range transport from Antarctic coastal emissions. Elevated CCN concentrations to the south of 60°S originate from biogenic emissions from the Antarctic coastal area."

l. 392 f: The information on the four regimes is repeated in l. 404ff. Consider removing some redundancy from the conclusions.

The sentence on line 404 defining the four regimes has been removed.

**Reviewer's comments copied from attached supplement document with responses:**

L. 23 "a location with elevated phytoplankton emissions relative to the rest of the SO." … There are regions in the Southern Ocean, e.g. around South Georgia, which are known for the productivity. This statement is too strong. Rather "relative to most other locations in the Southern Ocean".

We have updated this statement as follows:

> "In 5-day HYSPLIT back trajectories, air parcels with elevated CCN concentrations were almost always shown to have crossed the Antarctic coast, a location with elevated phytoplankton emissions relative to the rest of the SO in the region south of Australia."

L. 37 "The Antarctic coastal source of CCN from the south as well as CCN sources from the mid-latitudes create a latitudinal gradient in CCN concentration with an observed minimum in the SO between 55°S and 60°S."… Is the minimum only a question of sources? Between 55 and 60 °S many cyclones lead to precipitation. So the minimum can also be an effect of enhanced removal processes.

We agree and have updated the statement as follows:

> "The Antarctic coastal source of CCN from the south, CCN sources from the mid-latitudes and enhanced precipitation sink in the cyclonic circulation between the Ferrel and Polar cells (around 60 °S) create opposing latitudinal gradients in the CCN concentration with an observed minimum in the SO between 55°S and 60°S."

L. 51 "Schmale, 2019;"... incomplete citation.
Fixed

L. 54 A new publication on that topic is by Efraim et al. (2020): DOI : 10.1029/2020JD032409.
The citation has been added.

L. 70 "New particle formation (NPF) from the oxidation of marine biologically emitted VOCs occurs when the particle condensational 70 sink is low and temperature is low, both of which are prevalent conditions over the SO (Raes et al., 1997; Yue and Deepak, 1982)." … This is incorrect. The sea spray particles are large and actually produce a large condensation sink.

The text has been updated as follows:
> "New particle formation (NPF) from the oxidation of marine biologically emitted VOCs occurs mostly in the FT where the particle condensational sink and temperature are lower than in the MBL, which are prevalent conditions over the SO (Raes et al., 1997; Yue and Deepak, 1982)"

L. 72 "While new particle formation has been observed in the SO marine boundary layer (Covert et al., 1992; Humphries et al., 2015; Kyrö et al., 2013; Pirjola et al., 2000; Weller et al., 2015)" ... I am not familiar with all paper listed here, but the Weller et al. 2015 paper does not say that NPF occurred over the ocean. They report rather on NPF at a coastal Antarctic site which has very different characteristics from the open Southern Ocean.
See updated text in response to the next comment.

L. 75 " owing to the absence of PMA in the SO MBL"... This statement has to be revised or different literature needs to be cited. This does not make sense. Did the authors want to say owing to the absence of PMA in the SO free troposphere?

The text has been updated as follows:
> "While NPF has been observed in the marine boundary layer, often at coastal locations (Covert et al., 1992; Humphries et al., 2015; Kyrö et al., 2013; Pirjola et al., 2000; Weller et al., 2015), it occurs more commonly in the FT (Bates et al., 1998b; Clarke et al., 1998; Humphries et al., 2016; Odowd et al., 1997; Reus et al., 2000; Sanchez et al., 2018; Yoon and Brimblecombe, 2002) owing to the absence of PMA in the FT (McCoy et al., 2015)."

L. 81 "SO contains much less biological activity near the ocean surface relative to the Antarctic continental coast,"...This is only partly corrected. The area around South Georgia is highly productive. See https://doi.org/10.1002/jgrc.20270 for a climatology. So if measurements had been taken over that transect, the observations would be very different. Given the vast extent of the Southern Ocean it is important to consider regional variability.

We agree and have updated the statement to:
> "The remote mid-latitude SO south of Australia contains much less biological activity near the ocean surface relative to the Antarctic continental coast, which creates a latitudinal gradient in

aerosol concentrations driven by biogenic particle formation with the exception of some biological hotspots such as near South Georgia (Alroe et al., 2019; Humphries et al., 2016; Kim et al., 2019; O'shea et al., 2017; Odowd et al., 1997; Weller et al., 2018)."

L. 84 "This trend in biology is linked to observations showing a distinct transition in aerosol properties around 64°S where CN concentrations"... Again, this is a certain longitude band which is not necessarily representative of all SO. Be specific.

The statement has been updated as follows:
"Shipborne observations in the region south of Australia show a distinct increase in aerosol concentrations south of 64°S, where CN concentrations are about five times higher during the austral spring months" …

L. 85 "Regions of sea ice melt on the Antarctic coast have been observed to be a significant source of methanesulfonic acid (MSA) as well as DMS and organic nitrogen (Dallósto et al., 2017; Dowd et al., 1997; Vana et al., 2007),"... The organic nitrogen observations from the Dall'Osto (note spelling) paper have not been made in conjunction with NPF. We do not know if Norg plays a role over the Southern Ocean for NPF.

We have omitted organic nitrogen and the Dall'Osto et al., 2017 reference from this sentence.

L. 116 "In the event that the cloud droplets do not precipitate, the evaporated particles are larger than their original size since non-volatile compounds (i.e., sulfate and MSA) that condensed onto the cloud droplet remain in the particle phase." … For aqueous chemistry, one cannot really speak of condensation.

We agree and have updated the sentence to state:
"In the event that the cloud droplets do not precipitate, the evaporated particles are larger than their original size since aqueous oxidation of volatile compounds (i.e., DMS, MSA, SO2 and nitric acid) have formed non-volatile sulfates and nitrates that remain in the particle phase."

L. 118 "This added mass shifts Aitken mode particles to the accumulation mode (Hoppel et al., 1986; Hudson et al., 2015; Kaufman and Tanré, 1994; Sanchez et al., 2017)." … Schmale et al. (2019) describe this as well in their overview article. This would also be a good resource for measurements in sectors of the SO other than South of Australia and South of South America.

We agree that we should indicate that our measurements are representative of the region south of Australia; however, we have mentioned properties of other areas of the Southern Ocean as needed. We have added Schmale et al. (2019) as a reference in this statement.

L. 130 "In this study, we discuss airborne CN and CCN measurements from the SOCRATES campaign."… What about CAPRICORN?

We have added the following text to the end of the statement:
" and briefly discuss shipborne CCN measurements made on the R/V Investigator for the CAPRICORN2 campaign, which was conducted in the same timeframe and region as SOCRATES"

L. 143 "research flights (RF)"... inconvenient abbreviation, because it also means radiative forcing, and this is certainly the more common use. Might confuse readers.

We agree that acronyms can be confusing, but it is standard NCAR protocol to use RF for 'research flights'. We will keep RF in the manuscript.

L. 149 "deiced components"... This is not understandable to me.

We have updated the sentence as follows:
"Ambient subsaturated particles collected with the UHSAS were dried through a de-icing system (designed to vary the temperature and pressure of sampled air to prevent ice formation in the inlet)."

L. 150 The miniature version is not described in the 2005 reference. Either provide a relevant reference or describe the instrument.

The exact method described in Roberts and Nenes 2005 is used for the miniature CCN counter.

We have added the following sentence for clarity:
        "The miniature CCN counters are custom-made and operate with the same physical principles as described by Roberts and Nenes (2005). "

L.159 "hygroscopicity parameter ($\kappa_{70}$) " add information on how exactly this was done

The text has been updated and a sentence has been added to explain the calculation of the hygroscopicity parameter.

        "The CCN spectra and UHSAS number concentrations on the GV were used to estimate the hygroscopicity parameter at 0.07 µm diameter ($\kappa$70) for each MBL leg. For this calculation, the critical supersaturation is derived from the CCN spectra, where the UHSAS concentration at 0.07 um diameter is equivalent to the CCN concentration."

L. 171 "R/V Investigator CCN at 0.35% are analyzed and compared to the GV HIAPER CCN0.3 measurements." ... Why are they not interpolated to 0.3 % SS? Description of the inlet and position of the instruments on the ship is missing.

The R/V investigator CCN measurements have now been linearly interpolated to obtain an estimate of the 0.3% SS CCN concentration.

The following text has been added to address the ship inlet and position of instruments on:
        "Details of the aerosol sampling system on board the RV Investigator are presented in Humphries et al. (2019) and Alroe et al., (2020). In short, aerosol sampling occurred via a common sampling inlet mounted on a mast at the bow of the ship, located 18 m above sea level. The CCN counter sampled from a manifold located 8 m below the mast in the ships bow."

L. 177 "The average latitude, longitude and altitude (50-500 m) of the MBL legs on the GV HIAPER were used as starting points for the back trajectories." ... Unclear. To me this sounds that for each leg there was only one coordinate from which the trajectories were released? If this is the case, large

uncertainties are linked to this method in terms of meteorological properties obtained along the trajectories. This makes the results less trustworthy.

We have updated the text as follows:
  "The latitude, longitude and altitude (50-500 m) averaged for each CCN spectra (~150 seconds, ~15-20 km horizontal distance) collected during the MBL legs on the GV HIAPER were used as starting points for the back trajectories."

L. 193 "UHSAS particles in the SOCRATES campaign were not fully dried"...A quantification would be helpful to make this study comparable.

This has been answered in a previous comment.

L. 201 "The TEM analysis shows almost all particles > 0.2 µm in diameter consist of sea salt; and sea salt particles account for 25% to 100% of particle number concentrations at particle diameters > 0.4 µm (Saliba et al. submitted)." … This is inconsistent."almost all" means to me near 100 %. So how can only 25 % of particles > 0.4 um be sea salt?

This has been answered in a previous comment.

L. 207 "Figure 1b shows" … should reference 1a first.
Note figure 1 is now figure 2.
We believe Figure 2b should be referenced first, Figure 2 is designed so that figure 2a and 2c align with the y and x-axis respectively, of figure 2b and contain histograms of the data shown in figure 2b.

L. 212 "The regime thresholds were selected based 210 on the bimodality of observed CN and CCN0.3 concentrations shown by the histograms and kernel density functions in Figure 1a,c. Figure 1a,c also shows the kernel density estimate based on a normal kernel function.".... I only see one kernel density function. Why "also"?

The statement was repetitive, and the second sentence has been removed.

L. 213 "each measurement is represented by a normal distribution"... Why not log-normal?

A kernel density is a non-parametric (no assumptions involved) way to estimate a probability distribution. For the range of concentrations encountered here, there is also no compelling reason to represent measurements of concentration as log-normal.

L. 229 "the relative contribution of the accumulation-mode and the Aitken-mode are used to identify recent particle formation (RPF) events"... An explanation of how the Aitken-mode was estimated is missing.

This comment has been answered in the previous reference to L. 229.

---

## Author Response (AR2)

We thank the Editor and reviewers for taking the time to review the manuscript and provide their insight, which has allowed us to greatly improve the manuscript. We have responded to the new comments below in blue text.

**Editor Comments:**

Dear authors,

Many thanks for your revised manuscript.

Both reviewers and I have evaluated your revisions. Overall, the manuscript has greatly improved. Reviewer #2 raised the issue that a few technical details are still missing (see comments below). Concerning the mini-CCNC: Although you write that it follows the same physical principles as the standard CCNC, I miss some more technical details (e.g. on instrument flow and column properties (i.e. is is the same as for the standard one or are there differences?), etc.). It would be great if you could clarify on these aspects as well (you could add this to the supplement if needed).

Last but not least, please add to your manuscript the statement on data availability. The data should be stored on a public/open data repository (if possible with DOI), see guidelines here: https://www.atmospheric-chemistry-and-physics.net/policies/data_policy.html.

Thanks and kind regards

Paul.

Technical details are discussed below in response to the reviewer's comments. The data availability statement is now included.

**Reviewer 1 Comments:**

The authors have thoroughly answered all remarks and revised their manuscript accordingly. I have no further suggestions.

**Reviewer 2 Comments:**

N/A

**Reviewer 3 Comments:**

First of all, the manuscript has improved much. Thank you for addressing these issues.
Albeit this, I still have two minor issues that I hope the authors will address.

**1: CCN counter calibration (author response to 2 (b))**
I still have some concerns with respect to how Köhler theory is used to estimate the critical supersaturation. I will try to make myself clearer. When Köhler theory (saturation ratio = some water activity representation * Kelvin effect) is used, there are many models that can represent the water activity (E.g. see paper by Rose et al., 2008). If you are "assuming a water activity of 1.0" (as you now

state in the methods), then only the Kelvin term (curvature effect) is left and you cannot derived a critical supersaturation. I am sure that this is not your case.

On the other hand, if the "procedure for calibrating the miniature CCN counter is identical to the method described by Roberts and Nenes (2005)", then they assume a van't Hoff factor = 3 (stated in Roberts and Nenes, 2005). This water activity parameterisation has be shown to perform poorly compared to the E-AIM model (see Figure 11 (VH4b vs AP3), Rose et al, 2008).

To sum up, I miss the details on how the authors represent the water activity and estimate the theoretical critical supersaturation of monodisperse ammonium sulfate used for calibrating the CCN counter. Please address this issue.

In addition, I find it odd that there is no technical descriptions of this miniature CCN.

First, the statement of "assuming a water activity of 1.0" is not correct and was mistakenly included. This statement was actually intended for another manuscript that was being written at the same time as this manuscript. This statement has been removed from the text.

As the miniature CCN instrument follows the same principle as described in Roberts and Nenes, 2005 – albeit more optimized, technical details on instrument design have not been published. The flow rates and temperature gradients of the scanning CCN instrument vary from 0.09 lpm to 0.22 lpm and 8 K to 12 K, respectively. The length of the CCN wetted column surface is 138 mm. We used a sinusoidal pattern from high flow/high temperature gradient to low flow/low temperature gradient with a period of 10 minute to generate continuous CCN spectra every five minutes. We have updated the following text in the manuscript:

"An instrument model, discussed in Roberts and Nenes (2005) showed a standard deviation in the supersaturation estimate of about +/-0.01%. The supersaturation range of the scanning CCN counter flow rates and temperature gradients vary from 0.09 lpm to 0.22 lpm and 8 K to 12 K respectively. A sinusoidal pattern from high flow and high temperature gradient to low flow and low temperature gradient with a period of 10 minutes generated a continuous CCN spectra every 5 minutes that spanned from 0.06 % to 0.87 % supersaturation. The constant supersaturation CCN counter operated at constant flow and temperature gradient of 150 lpm and 9 K for a 0.43 % supersaturation (referred to as CCN0.43), at 1 Hz and was used to identify CCN gradients in vertical profiles (Section 3.6)."

In this manuscript, we used an upper limit (van't Hoff = 3). We recognize that the accepted values for kappa are 0.61 from CCN-derived values reported by Petters and Kreidenweis (2007) -- which lead to a van't Hoff factor of 2.52 (also reported in Rose et al., 2008). Using a van't Hoff factor of 2.52 implies that the CCN spectra would shift (< 10%) to larger supersaturation, and well within measurement uncertainties. There are no changes to the conclusions described in the manuscript. We have added the following text in the manuscript:

"The critical supersaturation in this study was derived by Kohler theory using a van't Hoff factor of 3.0 as an upper limit for ammonium sulfate. Using a van't Hoff factor of 2.52 (Petters and Kreidenweis, 2007; Rose et al., 2008), would shift the CCN spectra to larger supersaturations by less than 10%."

**2: Kappa derived from CCN spectra.**

You write now in lines 154-156: "The CCN spectra and UHSAS number concentrations on the GV were used to estimate the hygroscopicity parameter at 0.07 µm diameter (κ70) for each MBL leg. For this calculation, the critical supersaturation is derived from the CCN spectra, where the UHSAS concentration at 0.07 µm diameter is equivalent to the CCN concentration"

It is still not clear for me how to follow this calculation. Could the authors please clarify this to the reader?

E.g. in supplement, using an example or by pointing to another example elsewhere in litterature.

The following equation is used to derive the hygroscopicity parameter for ambient particles.

$$\kappa = \frac{4A^3}{27 D_{UHSAS}^3 (\ln S_c)^2}, A = \frac{4\sigma M_w}{RT\rho_w}$$

Where $\rho_w$ is the density of water, $M_w$ is the molar mass of water, $\sigma$ is the surface tension of water, $R$ is universal gas constant and $T$ is the absolute temperature (273.15 K), $D_{UHSAS}$ is the smallest diameter measured by the UHSAS (0.07 µm) and $S_c$ is the critical supersaturation in which the CCN concentration is equivalent to the total UHSAS concentration (see Figure S1 below for example). This equation is similar to that of Petters and Kreidenweis (2007), which uses the dry diameter at which the CCN reaches 50% of the total particle number concentration at a specified supersaturation as the activation diameter. Here, instead of determining the activation diameter from a specified supersaturation we are determining the critical supersaturation for a specified activation diameter (0.07 µm).

[Figure]

Figure S1: The UHSAS number distribution (left) is integrated to determine the number of particles greater than 0.07 µm (79 cm-3). Then the CCN critical supersaturation is determined with the CCN spectra (right), by identifying when the total UHSAS concentration was equivalent to the CCN concentration.

Reference:

Rose, D., Gunthe, S. S., Mikhailov, E., Frank, G. P., Dusek, U., Andreae, M. O., & Pöschl, U. (2008). Calibration and measurement uncertainties of a continuous-flow cloud condensation nuclei counter (DMT-CCNC): CCN activation of ammonium sulfate and sodium chloride aerosol particles in theory and experiment. Atmos. Chem. Phys., 8(5), 1153–1179. https://doi.org/10.5194/acp-8-1153-2008